# Regrafting submillimeter-scale ferromagnetic soft continuums

Yang Yang[1,3], Wentao Shi [1,3], Boguang Yang[2], Tiandi Xiong[2], Zhong Alan Li[2] & Hongliang Ren [1] ✉

Submillimeter-scale ferromagnetic soft continuums (FSCs) own innate skills in performing desirable and delicate bending for confined space navigation, especially in biological lumens. However, such tiny structures are difficult to endow with complex designs, thereby challenging to realize more sophisticated functions for various purposes, especially in vivo therapies and manipulations. Inspired by grafting for muscles and plants, we propose submillimeter-scale FSCs that can actively divide into pieces at any region, and conversely, the pieces can actively graft to each other to form the original structure or novel shapes. We define these functions as regrafting, comprising self-division and self-mergence. Implementing regrafting implies actively switching between two opposing characteristics: sufficient continuum structural strength for steering loads and a low fracture strength for division and mergence. Therefore, we developed ferromagnetic thermoplastic soft materials to replace the widely applied thermoset materials for continuums and shed the commonly required coating layers. Being made of the ferromagnetic material family that can undergo reversible elastomer-fluid transitions, the proposed FSCs can perform arbitrary division-mergence and navigate confined spaces for multiple endoscopic tasks in one go. Endowed with enhanced flexibility and reconfigurability in situ by regrafting, the proposed FSCs may open a multifunctional path for operating a wider range of biomedical tasks.

Submillimeter-scale ferromagnetic soft continuum (FSC) robots possess immense potential for executing intricate navigation, especially in confined and tortuous endoluminal scenarios[1–3]. Compared with conventional continuums driven by tendons and fluids, these FSCs offer enhanced steerability, remote controllability, and a significant reduction in overall size for adapting to narrow environments[4–6].

Efforts have been made to elevate FSC studies from the concepts to the endoluminal navigation stage. For example, a submillimeter-scale FSC was reported to navigate cerebrovascular networks and carotid arteries under external magnetic controls[1]. Fabricated by dispersing neodymium nanoparticles in a silicone rubber polymer matrix and coating it with a hydrogel layer, it can conduct accurate steerings and efficient navigation with reduced friction. This typical design was also reported to conduct delicate cell manipulations in intravascular environments[3,7].

However, a fundamental research gap hinders submillimeter-scale FSCs from realizing their advanced functionalities: Their tiny size and simple structure make it difficult to endow such continuums with complicated capabilities for a broader range of endoluminal manipulations. For example, the limited design space within these tiny continuums poses a dilemma for accommodating complex structures, thereby hindering them from performing basic tissue manipulations that traditional regular-size continuums can do, such as grasping[8,9], dissection[10], and thermal ablation[11]. Addressing this challenge would significantly advance the FSCs beyond the existing navigation-application stage to a new stage of in vivo manipulation and therapy.

[1]Department of Electronic Engineering, The Chinese University of Hong Kong, Hong Kong, SAR, China. [2]Department of Biomedical Engineering, The Chinese University of Hong Kong, Hong Kong, SAR, China. [3]These authors contributed equally: Yang Yang, Wentao Shi. ✉e-mail: hren@cuhk.edu.hk

To address the gap, we propose the regrafting mechanism and systematically investigate its fundamental principle. Inspired by grafting techniques for muscles and plants, known as transplanting tissues from one body part to another, we anticipate that the regrafting technique will endow similar functional/structural integration behavior to FSCs. Regrafting contains two functions: self-division and self-mergence. Regarding self-division, unlike the passive splitting of muscles and plants in response to external cuts, continuums with self-division ability are anticipated to actively divide into multiple sub-continuums throughout their structures. Self-mergence, beyond the commonly known self-healing techniques, can either restore the original form of divided sub-continuums or reorganize them into novel shapes to perform new functions. Therefore, a regraftable FSC is expected to actively split into several sub-continuums that can collaborate or independently execute tasks, and also merge with each other for various purposes. Regrafting is expected to significantly improve the functionality of FSCs while maintaining their tiny dimensions and simple structures.

Note that the proposed regrafting technique differs from the widely known kinematic reconfigurations by preprogramming mechanical mechanisms at specific points along the continuums[12]. We define regrafting as active division and mergence at any continuum region without the need for preprogramming. Achieving regrafting in submillimeter-scale FSCs is challenging: installing connection mechanisms will affect FSC flexibility and homogeneous nature[13]; current materials for FSCs face difficulties in actively dividing and merging.

Implementing regrafting requires the continuums to actively switch between two opposing characteristics: sufficient strength (e.g., ~MPa) for external loads and low fracture limit (e.g., ~kPa) for division and mergence. Existing FSCs face difficulties in realizing characteristic switching for the following reasons. First, typical FSCs are made of silicone-rubber-based thermoset materials[1–3,7]. Such materials are hyper-elastic and can only break at an extreme elongation (e.g., a strain of ~800%) under external loads. Additionally, they cannot be merged once the hyperelastomer is cured due to its thermoset nature. Second, a shell layer is commonly coated on FSCs for various purposes, such as lubrication[1,3,7], shape maintenance[14], and toxicity isolation[8]. For example, low-melting-point alloys (LMPAs) are regarded as characteristic-switchable continuum materials[8]. However, due to the excessively narrow temperature range for conducting solid-liquid transitions (e.g., several degrees Celsius), LMPAs exhibit two extreme conditions: excessively rigid stiffness in the solid state (~GPa) and a puddle of amorphous fluids without the coating layer. Therefore, implementing regrafting with existing techniques is challenging (see detailed motivation and creativity analysis in Supplementary Materials —Section 11).

We propose phase-changeable ferromagnetic elastomers, ETAMs (electromagnetically transitional and actuatable materials), specifically designed for soft continuums. Such materials are further developed into a ferromagnetic phase-change material family with a broader range of elasticity, more controllable stiffness, and better fabricability compared with our previous work[15]. By applying ETAMs to soft continuums, we propose a new type of FSCs that can perform regrafting, viz., self-division and self-mergence, due to their relatively wide phase transition temperature range ($\Delta T \approx 30\,^\circ C$, defined as the softening state). We call them electromagnetically transitional and actuatable material-based continuums (ETACs). The fused extrusion printing strategy is demonstrated to show the practical fabrication of our proposed continuums. Basic steerability and related design factors are experimentally and numerically studied to prove that the proposed continuum can operate navigation comparable to existing FSCs. We investigate the regrafting mechanism from the perspective of material, magnetics, and thermodynamics, and propose systematic strategies for division and mergence. We showcase the application values of

regrafting through biomedical demonstrations in confined channels such as the respiratory tract and the gastrointestinal (GI) tract. These demonstrations include using "continuum carriers" to send multiple sub-ETACs for in vivo missions, using "graftable end-effectors" for gentle airway foreign body removals, and applying "aided devices" for thermal ablations and submucosal dissections. These demonstrations highlight three fundamental properties of the proposed ETACs for various biomedical applications: (1) small size for navigating confined spaces, (2) sufficient flexibility for operational tasks, and 3) regrafting ability to create new structures/functions.

## Results

### Working principles of ETACs

ETACs, submillimeter/millimeter-scale FSCs (e.g., side lengths of 0.6 and 1 mm in this work), can perform magnetically induced bending motions for in vivo navigation. Aiming to solve complicated tasks in confined spaces (e.g., grasping, thermal ablation, submucosal dissection, etc.), ETACs can further operate regrafting through phase transitions induced by radio-frequency (RF) fields. These basic and advanced functions benefit from ETAMs, an elastomer-fluid phase change material.

ETAMs can perform reversible elastomer-fluid transitions, including the elastomer state (varying with additive proportions, i.e., ambient at 40 °C), the softening state (e.g., 40–70 °C), and the fluid state (e.g., >70 °C). ETAMs can perform elastic deformations at the elastomer state (Young's modulus of 1–80 MPa) and controllable flowing motions at the fluid state (~7 body lengths, BLs). By adjusting additive proportions, ETAMs' material properties can be correspondingly modified[15]. In this work, we divide ETAMs into two upgraded sub-families called N-ETAMs and F-ETAMs, which are, respectively, made of NdFeB magnetic nanoparticles (MNPs) with pure biocompatible thermoplastics (Polycaprolactone, PCL), and magnetite nanoparticles ($Fe_3O_4$ NPs) with PCL. Moreover, we prepared pure PCL for ETACs, referred to as P-regions. These sub-families serve different purposes: P-regions serve as the magnetically non-responsive parts. N-ETAMs exhibit dipolar responses to external magnetic fields, and F-ETAMs can efficiently generate heat for phase transitions under RF fields.

ETACs contain the P-regions main body, the N-ETAMs control region, and the F-ETAMs connection region, which are respectively called P-region, N-region, and F-region (Fig. 1a). P-regions are of moderately flexible stiffness with no magnetic particles, which have no RF responses for heating. N-regions contain permanent magnet particles and have the best flexibility among these three regions, showing less RF heating capability than F-regions, which contain soft magnetic particles for sensitive magnetic heating. Specifically, the P-regions are designed to provide a suitable stiffness range of 50–80 MPa. This allows them to effectively resist minor external load interferences while still being able to deform when bending motions are required. As the only magnetically non-responsive component, P-regions form the main body of the continuum and are regarded as the skeleton. F-regions exhibit soft magnetic behaviors and can effectively respond to RF fields. During RF heating, the magnetite particles within F-ETAMs generate hysteresis heat to soften the surrounding thermoplastic particles. N-regions, involving hard magnet particles, can perform dipolar magnetic responses once they are magnetized. Benefiting from the satisfactory controllability and flexibility (1–50 MPa), N-regions are regarded as the manipulation unit and are applied as the continuum tips. It should be noted that N-regions can also respond to induction heating, however, due to their hard magnet nature, the heating effects of N-regions are 50–70% lower than F-regions' (see detailed comparisons in the following sections). The aforementioned P, F, and N-regions can be connected by diverse sequences, constituting the multifunctional ETACs. Therefore, regrafting can be conducted at any region of the continuums without preprogramming once only N-regions and F-regions are involved.

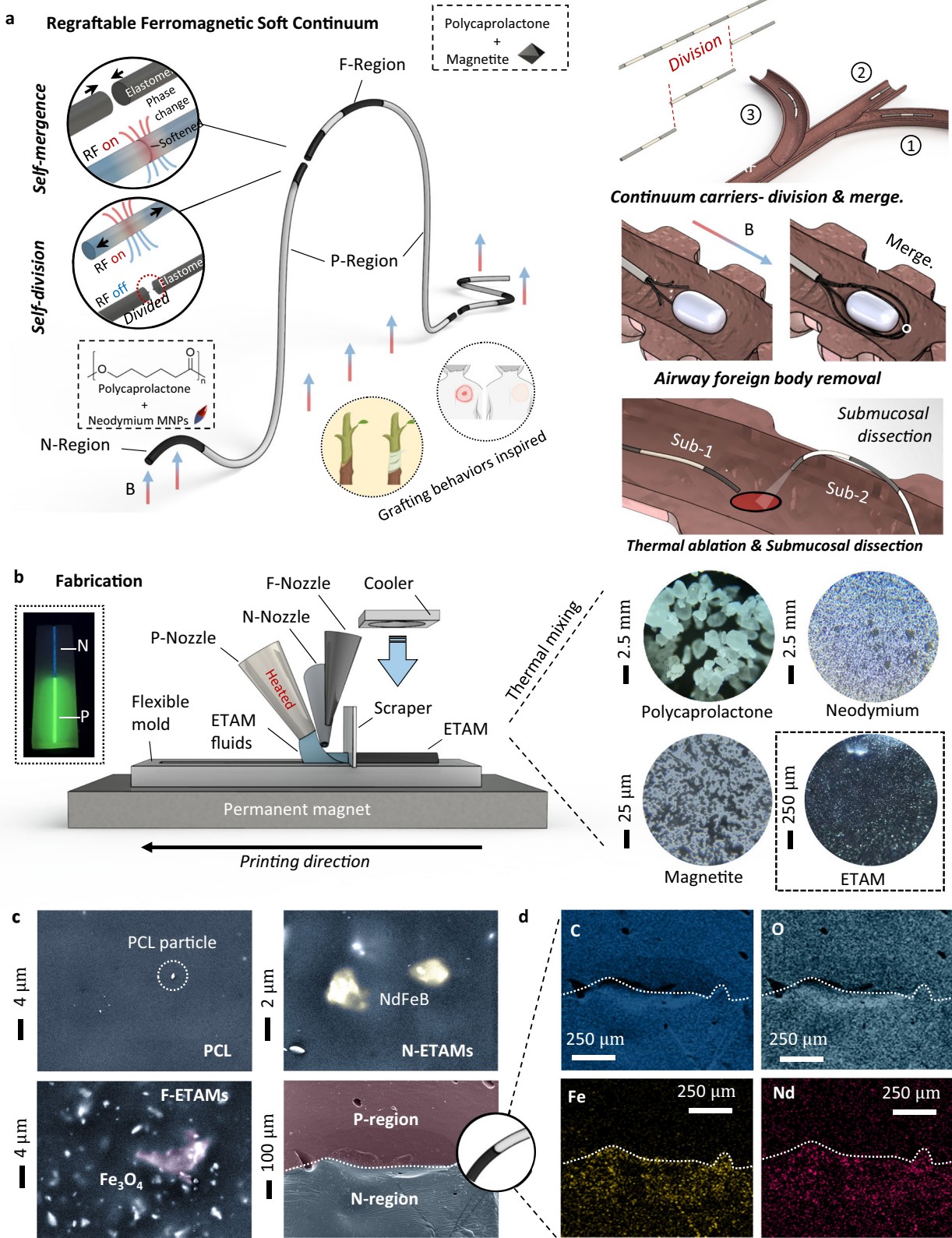

**Fig. 1 | Working principles and fabrication of electromagnetically transitional and actuatable material-based continuums (ETACs). a** ETACs respond to external magnetic fields and perform bending motions. At the same time, ETACs can also respond to RF fields operating regrafting. Therefore, ETACs are expected to play important roles in specific biomedical scenarios shown on the right. **b** ETACs can be easily fabricated using the fused extrusion process. The printable electromagnetically transitional and actuatable materials (ETAMs) can be prepared by thermal mixing raw materials, which are microscopically observed. **c** P-region, N-region, F-region, and P-N connection regions of the ETACs are observed using a scanning electron microscope (SEM). **d** The P-N connection region is further observed using element surface scanning analysis. See comparisons between N, P, and F are shown in Supplementary Movie S2. Created in BioRender. Yang (2025) https://BioRender.com/l415fio.

Regrafting contains two functions: self-division and self-mergence. ETACs can be separated into multiple sub-continuums at the F- and N-regions. Conversely, these sub-continuums can be merged to reconstitute the original shape or reconfigured into novel patterns, serving diverse objectives. All the aforementioned behaviors are anticipated to be executed actively.

Unlike LMPAs that perform excessively rapid phase change behaviors due to the relatively narrow temperature range of conducting phase transitions[16,17], ETAMs have an interphase between the elastomer and fluid states, called the softening state. In the softening state, F- and N-regions neither exhibit elasticity nor become amorphous puddles as liquids. F- and N-ETAMs can be divided by magnetic loads due to the reduced molecular attractions (i.e., the reduction of the yield point contributes to the increased ease of plastic deformation in the softened ETAMs). Conversely, multiple polymer matrices with reduced molecular attractions can easily adhere to each other. As the heat dissipates, the molecular attractions at the bond regions recover to their elastomeric properties (i.e., the yield point is enhanced back to its original state), enabling mergence.

Owing to the unique softening state, ETACs no longer require coated layers, thereby allowing them to exert their innate functionality without hindrance. As the new function of FSCs, Regrafting holds potential applications in complicated endoluminal tasks, such as divisions to operate tasks in multiple in vivo channels, foreign body removal, thermal ablation, and submucosal dissections (illustrated on the right panel of Fig. 1a).

## Fabrications and printable ETAMs

The thermoplastic nature of ETAMs determines the fabrication practicality of ETACs.

To achieve the one-time fabrication of ETACs, we have designed a printing platform (illustrated in Fig. 1b), which contains two units: a printing unit and a substrate unit. The printing unit comprises three nozzles for P-regions, N-ETAMs, and F-ETAMs, a scraper for geometry adjustments, and a cooler for rapid heat dissipation. The substrate unit consists of a silicone rubber mold with a straight-line groove and a permanent magnet under the mold to attract printed ETAMs, squeezing the hidden tiny air bubbles. Shortly after the printing (e.g., 1–5 min), ETACs can be easily demolded from the flexible mold and be magnetized for their normal uses. To further improve the fabrication practicality, we shaped the P-, N-, and F-ETAMs into glue strip shapes in batches. Inserting ETAMs-strips into hot adhesive guns allows ETACs to be easily fabricated manually (Supplementary Movie S1). It should be noted that the idea of using permanent magnets placed beneath the mold is to address the possibility (although low) that tiny bubbles existing around microparticles may expand during the printing process. To the best of our knowledge and experience in printing PCL parts, the commercially obtained PCL printing materials show less chance of generating obvious air bubbles during the printing process.

Fused extrusion printing is well-suited for ETAMs due to their satisfactory melt flow index range of 10–80 g/min. ETAMs can be prepared by mixing melted thermoplastics (150 μm) and magnetic nanoparticles (5 μm for magnetite and 48 μm for neodymium) at a temperature range of 160–200 degrees Celsius (Fig. 1b). Scanning electron microscopy of ETAMs revealed their compositions (Fig. 1c). Regarding N- and F-ETAMs, nanoparticles exist in an agglomerated form and are evenly dispersed within the polymer matrix. An elemental surface scanning analysis was conducted to locate the NdFeB and Fe3O4 clusters, as shown in Fig. 1c (Supplementary Materials—Section 1 and Section 2). The P-N connection region was observed: the boundary between P- and N-regions cannot be visually identified in SEM images, suggesting that the bonding between P-regions and N-ETAMs is strong (i.e., no obvious cracks exist; see detailed bonding test in the "self-mergence" section). Therefore, we further examined the P-N connection boundary according to iron and neodymium element distribution maps (Fig. 1d). It should be noted that the slight presence of iron and neodymium elements in the P-region is a result of fabrication imperfections, and no side effects were observed.

## Basic performances

Steerability is an essential function of a FSC, especially for in vivo navigation. We considered multiple factors for designing endoluminal continuums, including material types, magnetization properties, length-width (i.e., axial length vs. side length) ratios, and graded stiffness (Fig. 2a).

Elasticity and magnetization capability are fundamental in designing a soft continuum obtained by stretching and vibrating sample magnetometer (VSM) tests, respectively. Results show that different amounts of MNPs in ETAMs yield a modulus range of 1–80 MPa. Results also reveal that the magnetizability positively correlates with the MNP mass fraction (see detailed results in Supplementary Materials—Section 3).

The thermoplastic nature of ETAMs offers unique advantages regarding thermal effects on bending and modulus. When ETAMs are heated to 60 °C for more than 15 min, the molecular forces between PCL particles are significantly weakened, resulting in a softening phenomenon. The material takes approximately 9–10 h to recover its original stiffness. To modify such characteristics, we mixed the original PCL powder with another type of PCL powder (~10 times more rigid than the original PCL; see detailed techniques in Supplementary Materials—Section 4). The mixed PCL with flexible-rigid powder ratios of "2:1", "6:1", and "1:0 (Pure)" were tested to evaluate their time- and temperature-dependent steering performances and stiffness (Fig. 2b, c, respectively). Results show that ETACs benefit from the thermal effects with a stiffness softening rate of more than 1000%, resulting in significant steering performance enhancements (~500–1000% increment of the maximum bending angles). The softening effect of preheating is explained in detail in Supplementary Materials S23.

The length-width ratio effects were investigated by actuating ETACs within a Helmholtz coil (Fig. 2d). Various combinations of widths (0.6 mm and 1 mm) and tip lengths (30, 60, and 90 mm) were assessed. Regarding the width effect, although a width increment can allow a greater mass of MNPs for a stronger magnetic response capacity, the width reduction dominates the bending behavior due to the reduced cross-sectional area (maximum bending angle of the 0.6 mm ETAC is 10-20% higher than that of the 1 mm ETAC). Further, with the tip length increment, the width-domination effect becomes more pronounced (500% enhancement at maximum).

In practical tests, connecting the flexible tip materials (1–50 MPa) to the rigid P-regions (more than 50 MPa), resulted in connection failures and inefficient stress transitions due to the stiffness discontinuity (Supplementary Movie S2). Therefore, we designed a stiffness gradient structure at the tip region (see detailed methodology in Supplementary Material—Section 4).

The tip was evenly triple-divided into the soft ETAMs ($N_S$), transition ETAMs ($N_T$), and rigid ETAMs region ($N_R$). The bending effects were experimentally tested in electromagnetic and external permanent magnetic fields. The experimental results were further validated by numerical simulations (Fig. 2e, see detailed model setups in Supplementary Materials—Section 5). The results reveal that the overall bending effect is most sensitive to the stiffness of $N_R$. To analyze the relationship between the maximum bending angle $\theta_{Max}$ and the magnetic flux density $B$, parametric studies were conducted within the numerical model by varying one stiffness while keeping the other two stiffnesses consistent (Fig. 2f). The results indicate that the stiffness of $N_R$ is crucial in balancing a smooth stiffness transition with P regions and allowing for an enhanced range of motions (see solid lines in Fig. 2e). The stiffness of $N_T$, plays a role in facilitating a smooth

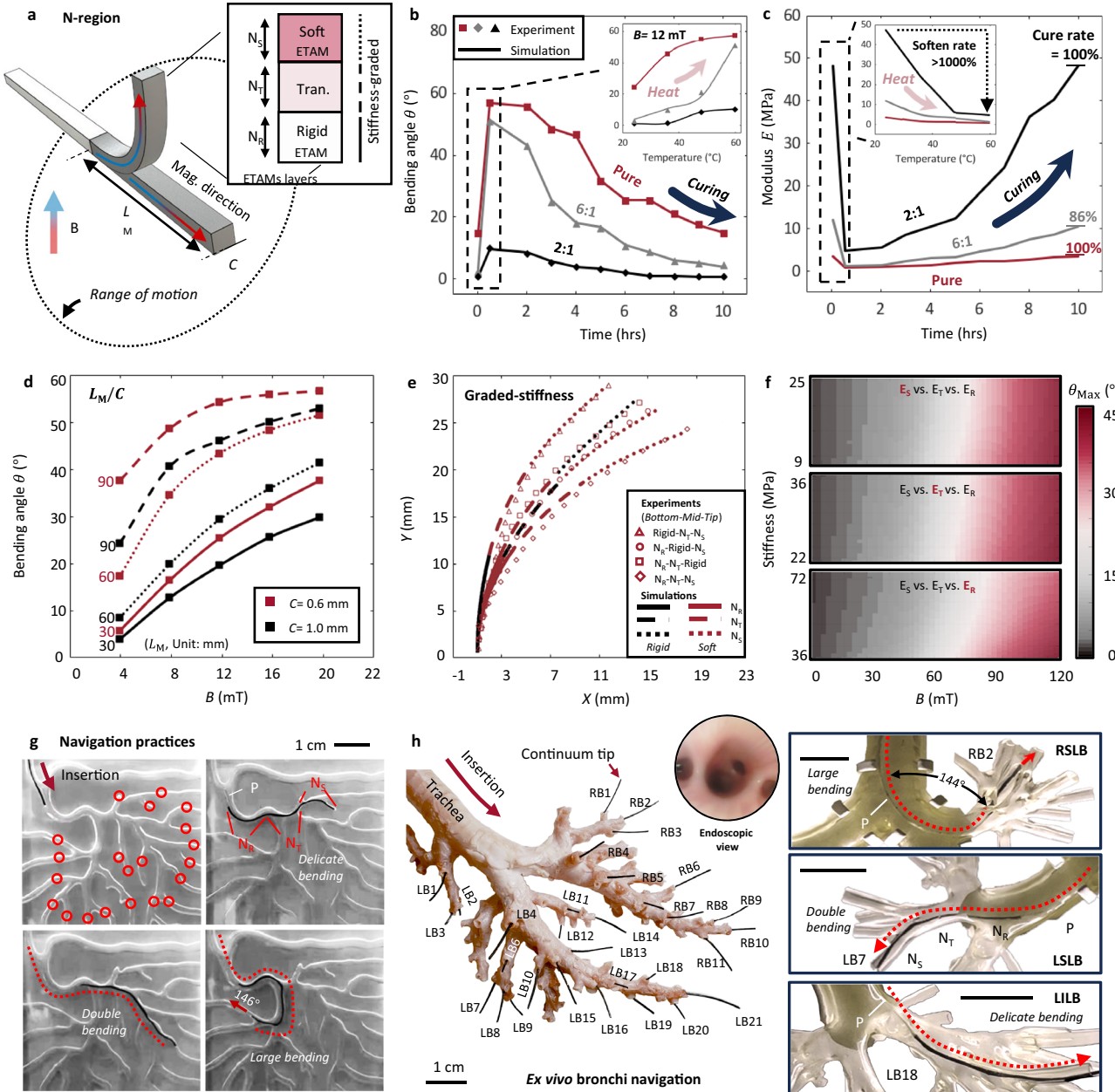

**Fig. 2 | Basic bending performances and navigation of ETACs (electromagnetically transitional and actuatable material-based continuums). a** Design principle of the ETACs. Design factors, including the tip length-width ratios and graded stiffness distribution, were considered. **b** Thermal effects on bending performances. Two series of tests were conducted regarding the thermal effects. The plot legends "2:1", "6:1", and "Pure" denote the mass ratio between flexible and rigid PCL (Polycaprolactone) powders, where "Pure" refers to the case without using rigid PCL powders. **c** Thermal effects on stiffnesses. The time-dependent and temperature-dependent stiffnesses are presented. It will take ~9.5 h to complete a cure cycle after being heated by 60 °C for 30 min. The softening rate of ETACs is larger than 1000% after being heated. The test was conducted in Helmholtz coils with 12 mT magnetic field strength. The modulus was calculated by the validated numerical model. **d** Bending performances of ETACs with different length-width ratios were tested in a Helmholtz coil. **e** Graded stiffness at ETACs' tip was tested with a constant length-width ratio ($\frac{L_M}{C}$ = 30) in permanent magnetic (PM) fields at a

distance of 4 cm. Experimental and numerical results were compared with agreements. **f** The validated numerical model was further applied to calculate the maximum bending effect of ETACs with respect to magnetic flux density and material stiffness. **g** Navigation practices. ETACs were applied to navigate a complex aerodigestive tract under PM fields. ETACs with and without optic fiber cores successfully reached all predefined targets framed in circles. **h** An ETAC navigated through an ex vivo pig bronchi system. Inserted from the trachea, the continuums successfully reached 32 predefined targets at different bronchus levels under PM controls. The right-side images show the continuums' deformation status during their navigation at RB2 (right superior lobar bronchus, RSLB), LB7 (left superior lobar bronchus, LSLB), and LB18 (left inferior lobar bronchus, LILB), respectively. In (**g**, **h**), the graded stiffness was given to the tested continuum tip pre-heated in a 50 °C oven for 30 min and cooled down to ambient temperature for 5 min. Scale bar: 1 cm. Source data are provided as a Source Data file.

transition of bending curvatures between $N_S$ and $N_R$, ensuring the continuous steering shapes (see dashed lines in Fig. 2e).

Featuring graded stiffness at the tip, ETACs exhibit satisfactory steerability (i.e., large bending of ~145°, multiple bending with more

than 4 curves, and delicate bending within 5-60°) for narrow channel navigation. We tested both ETACs with and without an optical fiber core in a complex aerodigestive tract (Fig. 2g). ETACs successfully reached all predefined targets with challenging bending

performances, where the graded stiffness design played a significant role in resisting minor external load interferences and adapting to rugged terrains (Supplementary Movies S3, S13–15.). It should be noted that ETACs can complete navigation with both external permanent magnet control and electromagnetic control (Helmholtz coils). The permanent magnet can control ETACs from a distance of more than 10 cm for practical uses. Details can be found in Supplementary Materials S8, S9, and S14.

To maintain ETACs' simple structural design and regrafting function, we applied medical device lubricants (No. LUB0005, Health&Beyond Hygienic Product Inc., China) as instructed by medical doctors, instead of using lubrication coating layers. The results show that the lubrication did not improve a uniform modulus ETAC's navigation performance. The uniform modulus ETAC failed to complete a ~270° steering due to the unoptimized force transmission, which is not directly related to lubrication conditions. When the graded stiffness structure is applied to ETAC's tip, no obvious differences were observed between ETACs with and without lubrication.

An ex vivo pig bronchi system was utilized to further showcase the navigation ability of ETACs (Fig. 2h). Inserted from the trachea, the continuum successfully reached all 32 predefined bronchi at different bronchus levels (Supplementary Movies S4, 5), such as the main bronchus (MB), superior lobar bronchus (SLB), middle lobar bronchus (MLB), and inferior lobar bronchus (ILB). Additionally, a bronchi phantom was used to provide a clear visualization of the bending status during navigation to different bronchi levels. For example, the continuum performed a large overall bending (RSLB, ~144°), double bending at the RB2, LB7 (LSLB, +33° and −36°), and a delicate overall bending at LB18 (LILB, 43°), respectively. See the lubrication needs explanation in Supplementary Material S16.

## Self-divisions

Untethered self-divisions can be conducted at both F-regions and N-regions due to their RF responsiveness (Fig. 3a). An entire magnetized ETAC can be divided into several sub-continuums while maintaining the original magnetization profiles. Self-division implementation strongly depends on a reliable strategy to stably keep the continuums' temperature in the softening range. There are several determining factors, such as materials, coil shapes, heating distances, heating positions, etc. A good understanding of the influencing mechanism will enhance the potential for practical applications, especially in endoluminal scenarios. For example (Fig. 3a), disk-shaped coils could be applied for confined remote spaces (e.g., respiratory tract and GI tract).

Thermodynamic performances of ETAMs under RF heating were investigated by infrared (IR) imaging analysis (Fig. 3b). An F-ETAM specimen completed an elastomer-fluid phase change in less than 20 s. Adjusting the material type, heating distance, and induction coil shape can slow down the heating process: an N-ETAM specimen was just softened within the same time (20 s). Quantitative analysis of temperature-time relations was conducted by tracking specimens' surface temperatures (Fig. 3c). The F-ETAMs with an 80% NP fraction (calculated as the mass ratio between NPs and base materials) served as the control group. The temperature rapidly reached ~120 degrees Celsius in ~10 s, causing the material to go through the elastomer, softening state, and fluid state. The heating speed can be further accelerated by increasing NP fractions, shown as the F-100% group. We replaced the magnetite NPs with neodymium MNPs to slow the heating process, which took twice as long to reach the same temperature as the control group (~30 s to 120 degrees Celsius). Another approach we tried was to increase the heating distance. Like other RF-enabled robots, heating performances will be significantly suppressed with the increment of the heating distance[18–20]. With a doubled heating distance compared to the control group, the heating time consumption was more than twice as slow. Further, with a tripled heating distance, the

heating performance was significantly suppressed[21], which took ~1 min to obtain a 10 degrees Celsius increment. Additionally, the shapes of induction coil directly determine the conditions of magnetic field concentration. We compared two commercially available induction coils: a solenoid (called C1, in Fig. 3a) and a disk (called C2, in Fig. 3a). According to the results, the heating efficiency of the disk-shaped coil is visibly lower than that of the solenoid-shaped coil. This is because the former generates axially superposed RF fields, while the latter generates superposed fields at the semispherical center. Overall, by varying material types, heating distances, and coil shapes, a ready-to-use library of ETAMs' thermodynamic performance can be built to meet various application requirements.

After ensuring that ETAMs' thermodynamic performance is controllable and stable, we conducted IR imaging on ETACs under local RF heating for self-divisions (Fig. 3d, Supplementary Movie S6, S16, and S20). An N-P-F-P-N continuum was applied to conduct the local heating test from right to left (Supplementary Movie S7). We note that the applied sequence of ETAM regions is "N-P-F-P-N" for control experiments. However, in other cases, there is no need to preprogram ETACs in this order. For example, an ETAC designed only with an F- or N-region can also complete tetherless self-divisions, which will split at RF-concentrated regions. Within a heating time of 10 seconds, the N-region was softened precisely with no effect on its neighboring regions (Fig. 3d-i&iv). Compared with the N-region, the F-region showed higher heating efficiency (Fig. 3d-ii, ~10 degrees Celsius higher than the N-region in the same heating duration). At the same time, the heated N-region performed a ~8 degrees Celsius cooling-down behavior due to the heat dissipation. Moreover, no thermal responses were observed as expected under another 10 s local heating at the P-region (Fig. 3d-iii).

The abovementioned facts prove that local heating is accurate and targeted to the aiming region, without affecting the hardening process of neighboring regions. This phenomenon enables ETACs to conduct accurate self-divisions necessary for practical applications.

Therefore, we applied ETACs to demonstrate the self-division function using different division strategies (Fig. 3e). The continuum was magnetized in the axial direction, and its F-region was chosen as the division region (Fig. 3e–i). There are three typical division strategies: the pulling division, twisting division, and bending division. These strategies can be specifically selected for different working scenarios. The division by pulling is suitable for radial-limited scenarios, which can be realized by attracting the remote tip and pulling the insertion end, either magnetically (i.e., untethered) or manually (i.e., tethered), to divide the softened F-region (Fig. 3e-ii). When it comes to scenarios with both radial and axial limitations, the twisting division can be adopted. The softened F-region can be easily divided by attracting the remote end and twisting the insertion end due to the stored strain energy at the twisting region (Fig. 3e-iii). Moreover, the bending division strategy can be particularly applied to forked terrains. Unlike the other two strategies, the bending division is realized by holding the insertion end and bending the remote end with magnetic fields to break the softened F-region (Fig. 3e-iv). After self-divisions, sub-continuums can still respond to external magnetic fields, generating corresponding bending performances. It should be noted that the self-division can be realized both manually and magnetically. The manual approach is simple and direct for use in relatively large lumens, while the magnetic approach shows apparent advantages in completing self-divisions in complex and confined lumens.

During the self-division process, a typical consideration is whether the magnetization profiles of N-regions become randomized due to temperature elevations and phase changes. Therefore, a quantitative analysis was conducted on the magnetization profiles of ETACs (Fig. 3f). The continuum's magnetization profile was measured and recorded by a magnetometer and compared with the profiles after being heated at different temperatures. Results showed that during

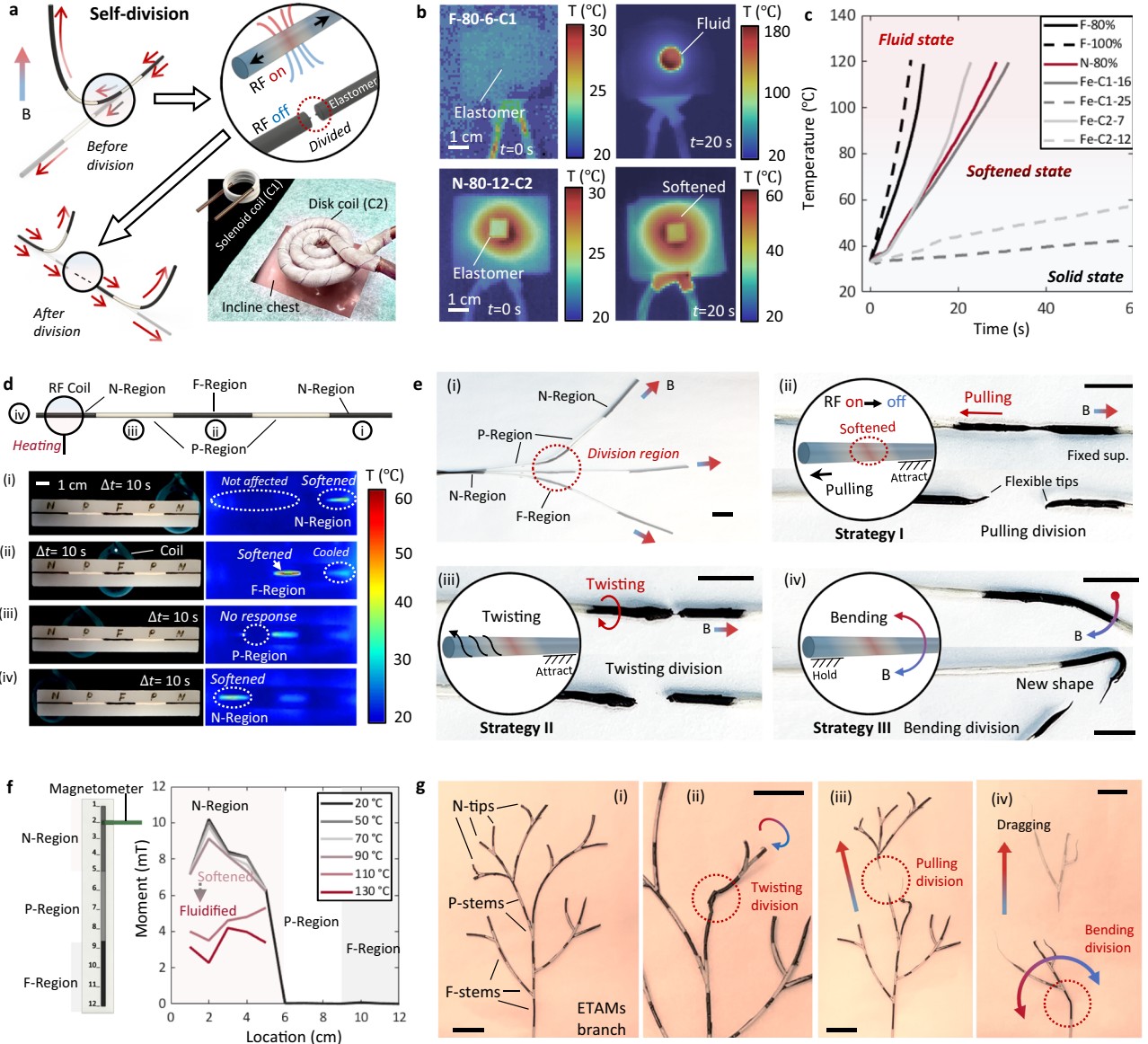

**Fig. 3 | Self-divisions of untethered ETACs (electromagnetically transitional and actuatable material-based continuums). a** Illustration of the self-division working principle. An entire ETAC with a predefined magnetization profile can be divided into several sub-continuums while maintaining their magnetization profiles. Considering the practical implementation in biomedical scenarios, customized induction coils can be applied to conduct different surgical tasks. **b** Infrared images showing the ETAMs' (electromagnetically transitional and actuatable material-based materials) thermodynamic performance under the RF heating for self-divisions. The number on the top-left side denotes the applied material type: nanoparticle (NP) type-NP weight fractions (%)-distance to the coil (mm)-induction coil types. The NP weight fraction is calculated as the ratio of NP mass to PCL (Polycaprolactone) mass. C1 and C2 in the figure legend refer to the solenoid and disk-shaped coils, respectively. **c** Time-dependent temperature changes during RF heating with respect to material types, distances to coils, and coil types. Cases without showing NP weight fractions, coil types, and heating distances are regarded as default values of 80%, C1, and 8 mm, respectively. **d** Infrared and real-time images of ETACs' thermodynamic performances under the RF heating generated by a solenoid induction coil. Numbers of "i"-"iv" denote the heating priority. **e** Self-division strategies. (i) An entire ETAC can respond to external magnetic fields performing bending motions. The division region is preset at the F-region in the middle of the continuum. (ii) Pulling division strategy. (iii) Twisting division strategy. (iv) Bending division strategy. Scale bar: 1 cm. **f** Relations between the heating temperature and maintained magnetization profiles. Results show that divisions in the softening state have no obvious negative effect on maintaining magnetization profiles. **g** An ETAMs-branch system withers its branches by different self-division strategies. Scale bar: 1 cm. Source data are provided as a Source Data file.

the elastomer and softening states (i.e., 20–70 °C), both the magnetization directions and magnitudes remained the same, indicating that the magnetization profiles do not become randomized when conducting self-divisions on N-regions. Significant changes in magnetization distributions and values were observed as the temperature was further increased, shifting the softening state to the fluid state. Maintaining ETACs in the softening state is crucial to realizing self-divisions. Magnetic particles can rotate under an external magnetic field rather than transmitting torques when the PCL matrix is significantly

softened. However, this phenomenon was not observed in our case because the selected PCL in the softening conditions still had sufficient stiffness to prevent changes in magnetization (Supplementary Materials S19).

We demonstrated the extension of the untethered self-division function (Supplementary Movie S8). ETACs can be fabricated into an ETAMs branch system containing multiple N-tips, F-stems, and P-stems. The branch system can separate sub-branches using different division strategies (Fig. 3g). This demonstration also addresses how to

assemble and apply a fiber-equipped continuum and a pure ETAC simultaneously: The fiber-equipped continuum can be regarded as a trunk and assembled with the sub-stems made of pure ETACs. We also note that all the abovementioned tetherless manipulation strategies can be completed remotely without applying loads to continuums, as shown in this demo. To further showcase the application value, we utilized the endoscope-equipped ETAC branch system to conduct in vivo object grasping and releasing within a complex terrain (Supplementary Movie S9). To effectively heat the target region, we provided detailed engineering techniques to reduce the distance effect on the heating area (Supplementary Materials S18). The inherent mobility characteristics of the divided sub-ETACs facilitate the active division and active motions to desired locations before merging with the ETAC main body. Therefore, the mobility was studied in detail, as shown in Supplementary Material S25.

## Self-mergence

As the reverse function of self-divisions, with both the untethered and tethered manipulation strategies, self-mergences can be regarded as another key step of the regrafting after divisions (Fig. 4a). Self-mergences can be conducted between a magnetically responsive region and any other regions, such as F-F, F-N, F-P, and N-P. By maintaining the continuums' original magnetization profiles and combining their structure patterns, self-mergences significantly broaden ETACs' functional flexibility. Three main strategies for implementing self-mergences are face-to-face mergence, line-to-line mergence, and point-to-point mergence.

Face-to-face mergence refers to the process of reconnecting sub-continuums by bring their radial sections into contact (see Face-to-Face principle in the right panel of Fig. 4a) and operating phase changes at the region (Fig. 4c-i&ii). Similar to self-divisions, RF heating can be applied to the contacted region stably and controllably (i.e., heating effects concentrate at the circular center region of the RF coils[21]). With an axial magnetization direction from left to right (see direction in the left panel of Fig. 4c-ii), two sub-continuums were merged and performed an overall dipolar bending (see the zoomed-in performances in the right panel of Fig. 4c-ii) under a unidirectional magnetic field (Fig. 4c-ii). The sub-continuums' thermal response during the mergence was captured by an infrared camera (Fig. 4c-v).

Line-to-line mergence refers to the side-wall connection of multiple sub-continuums (see Line-to-Line principle in the right panel of Fig. 4a). It should be noted that the $N_1$ and $N_2$ bend in different directions under the same field (Fig. 4d-i) and apply forces to firmly align $F_1$ and $F_2$ for mergences. Side boundaries of $F_1$ and $F_2$ can bond to each other by applying induction heating, which results in an overall dipolar bending performance under a unidirectional magnetic field (see the detailed direction in Fig. 4d-ii).

Point-to-point mergence refers to the connection between cross-contacted regions of different sub-continuums (see Point-to-Point principle in the right panel of Fig. 4a). Dipolar bending motions were also observed after the mergence (Fig. 4e-ii). Compared with the other two strategies, the point-to-point mergence took longer (~3 min for point-to-point and ~1 min for others) due to reduced contact areas (i.e., only $0.6 \times 0.6$ mm² for the point-to-point case). Overall, mergences using different strategies can be completed in 1–3 min. Note that, like self-divisions, the time required for mergence also depends on multiple factors, i.e., materials, heating distances, and coil shapes.

The alignment issue reveals another advantage of ETAC: conducting self-alignment due to the continuum's nature.

Because of the continuous magnetization distribution, the divided segments can be regarded as multiple magnets that maintain their original magnetization profiles. Regardless of the division methods adopted or the geometries of the division region, by bringing the two division ends close to each other, the divided segments can attract and align with each other (Supplementary Materials S21 and Supplementary Movie S19).

The deformation prediction after ETACs' self-mergence is convenient and straightforward, which is an advantage of ETAC's simple structure design. F-regions are used for division and mergence tasks, while N-regions are employed to conduct deformation tasks, due to the RF responsive capability of F-regions and the dipolar magnetic performances of N-regions. When F and N-regions are simultaneously under an external magnetic field, F regions tend to be attracted in the gradient magnetic field direction, while N-regions exhibit dipolar behavior with respect to the field direction. These deformations are directly predicted by numerical simulations (Supplementary Materials S22).

## Applications

Regrafting is expected to enable the submillimeter/millimeter-scale FSCs perform complicated tasks in confined spaces, thereby contributing to in vivo manipulation and therapy. Here, we provide three demonstrations to showcase ETACs' potential in biomedical applications: (1) Continuum carriers can be divided into several pieces to conduct different tasks and merged back into a single continuum to simultaneously grasp multiple objects out of the channel. (2) Graftable end effector can remove airway foreign bodies with different properties. (3) ETACs have the potential to assist an electrosurgical unit in performing submucosal dissection.

Submillimeter-scale FSCs can conduct narrow channel navigation. Given the enhanced flexibility and functionality of regrafting, ETACs are expected to conduct in situ self-divisions into several sub-continuums, operating their tasks in different subchannels. After completing their individual tasks, they can be merged in situ and retrieve multiple objects simultaneously (Fig. 5a). This process is similar to aircraft carriers sending and receiving planes. Therefore, we named this application "continuums carriers".

A continuums carrier, in this case, is predesigned with a structure of $N_1$-$P_1$-$N_2$-$P_2$-$N_3$-$P_3$-$F_4$-$P_4$ (NP4FP-ETAC), which can be divided into three N-P and one F-P sub-continuums. The N-P sub-continuums are separately applied to conduct delicate grasping in multiple subchannels. The F-P continuum is used as the insertion end and mergence region for the three N-P sub-continuums. We note that the NP4FP-ETAC is predesigned to better demonstrate the continuum carrier application. This can also be realized by an N-ETAC without programming (see Fig. 5a).

Three complicated tasks are expected to be accomplished in situ and tetherlessly: (1) Grasping an irregularly shaped object with a narrow hole (i.e., 1.5 mm diameter); (2) Accurately aiming, contacting, and sticking to a target (i.e., the circular center of a target with a radius of 1 mm); (3) Circumferentially grasping a rotary body with a significant local bending (i.e., a 2 cm-long tip bends more than 200°).

Three steps were adopted to accomplish the entire tetherless mission in situ: divisions and insertions, phase changes and grasps, and mergences for simultaneous grabbing-out (Supplementary Movie S10).

Divisions and insertions: the ETAC was inserted into the main channel guided by the $N_1$ tip (Fig. 5b-i). The tip can be further inserted through the narrow hole of an irregular object in subchannel-1 (Fig. 5b-ii). Self-division at the $P_1$-$N_2$ region was realized by locally applying an RF field to release the sub-continuum-1. Repeating the same approach, sub-continuums 2&3 were inserted and released into subchannels 2&3, respectively (Fig. 5b-iii&iv).

Phase changes and grasps: The sub-continuum-1 bent above the irregular object to form a hook shape (Fig. 5c). An RF field was applied at the tip region to soften the elastomer, enhancing the continuum's flexibility to adapt and remember the hook shape. The shape memory effect was realized by phase changes that eliminated the strain energy. The sub-continuum-2 was accurately aimed at the predefined target by

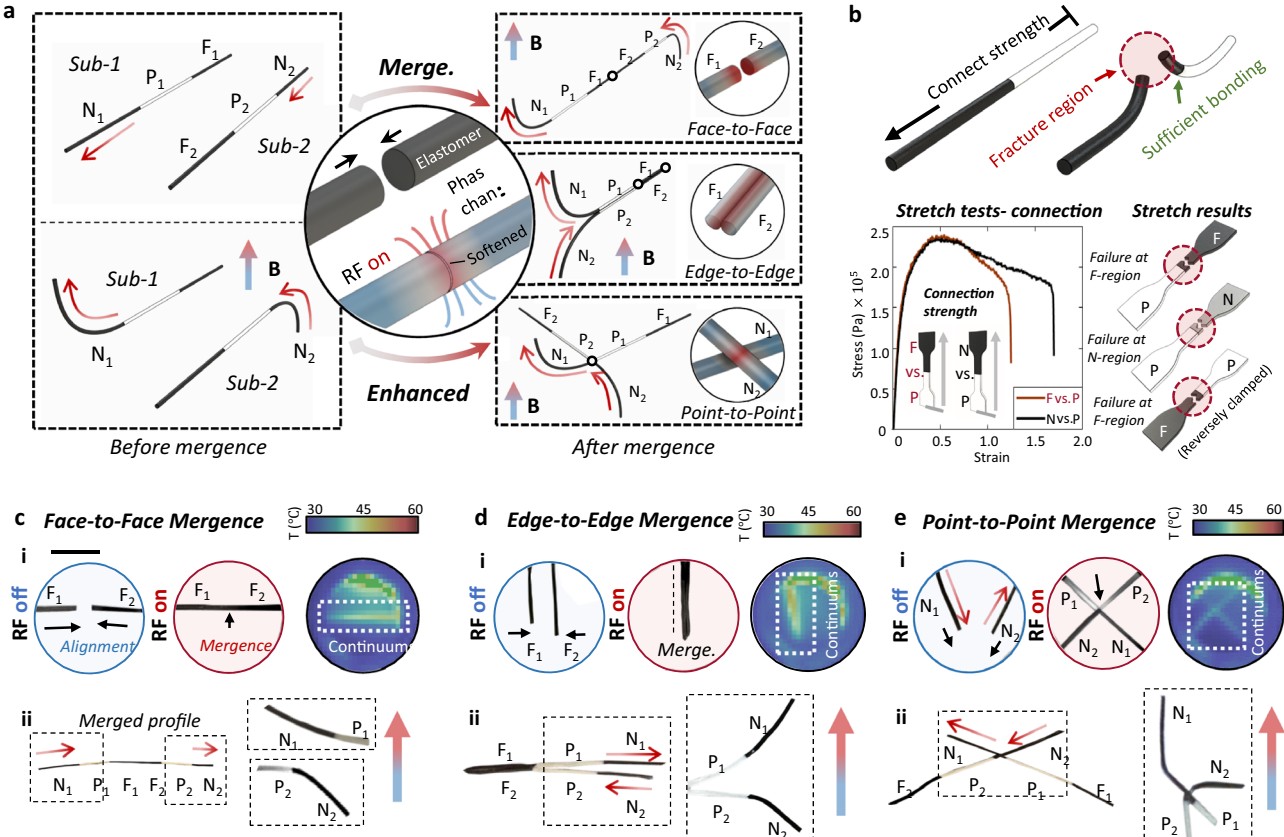

**Fig. 4 | Self-mergence of untethered ETACs (electromagnetically transitional and actuatable material-based continuums). a** Illustration of self-mergence principles. Self-mergence can be implemented by different mergence strategies: face-to-face, line-to-line, and point-to-point. N, P, F with subscripts denote the ETAMs (electromagnetically transitional and actuatable material-based materials) regions of corresponding sub-continuums. **b** Stretching tests of self-merged ETAM samples. Failures happened at the F/P regions regardless of the stretching direction and material types, suggesting that the bonding strength between P and F/N is sufficient to take external loads. **c–e** Real-time images show self-mergence processes enabled by different strategies that realize various enhanced functions. Infrared imaging was conducted to track the thermodynamic performances during the self-mergence processes. Scale bar: 1 cm. See temperature monitoring in Supplementary Material S15. See further descriptions in Supplementary Material S25. Source data are provided as a Source Data file.

magnetic controls and delicately bonded to the object by phase change (Fig. 5d). The bonding temperature is acceptable for the human body (45–50 °C for ~10 s). Note that the bonding process was conducted without contact with the channel's inner wall. The sub-continuum-3 circumferentially grasped the rotary object by a local bending at the $N_3$ tip. However, due to the extremely large local bending requirement (i.e., horizontally covering the cylinder with >200° bending angles and ~1.5 mm radius of curvature), the $N_3$ tip exhibited insufficient stiffness. Applying an RF field in this region enhanced the tip flexibility, allowing it to achieve a bending angle of more than 200 degrees (Fig. 5e).

Mergences and simultaneous grabbing-out: the $F_4$-$P_4$ sub-continuum was guided to the insertion ends of the other N-P continuums. After a mergence between $P_1$, $P_2$, $P_3$, and $F_4$, these sub-continuums were reassembled into an entire continuum, which can be pulled out to simultaneously extract all the objects (Fig. 5f). Note that the $N_1$-$P_1$-$N_2$-$P_2$-$N_3$-$P_3$-$F_4$-$P_4$ sequence is designed for a better illustration and is not the only choice to complete the tasks (i.e., ETACs can complete the task without preprogramming).

Airway foreign body removal has been a challenging procedure studied since the late 1800s[22–24]. The airway retrieval basket is a commonly used medical device designed to enclose the foreign body for physical removal[25–27]. However, aspects such as patient experience, object capture precision, and adaptability to various shapes are still under discussion[28]. For example, the foreign body should be physically constrained in the basket, but there is a probability that the object could be disturbed and escape. Another consideration is that, although current retrieval baskets can adapt to various foreign body shapes due to their soft structures, their performances in capturing foreign bodies with challenging shapes should be further improved (e.g., stick- and strip-shapes). Given the ETACs' flexibility and functionality, we have designed and applied a graftable end-effector to provide a potential solution to the above issues (Fig. 6a).

A graftable end-effector contains a P-root and four N-stems (premagnetized) with F-tips. The end-effector can respond to external magnetic fields, performing basic functions such as opening and grasping. Further, F-tips can bond to each other by self-mergences, providing sufficient restriction force to remove the object. Reversely, the bonded tips can unlock each other by self-divisions (Fig. 6c).

A nut (with a diameter of ~5 mm in this case), a common object that can accidentally enter the airway, was applied to showcase the working process of the graftable end-effector.

The end-effector was inserted through the trachea and positioned in the working region (Fig. 6d-i). To avoid disturbing the object, a magnetic field was applied to open the N-stems (Fig. 6d-ii). As the end-effector continues to insert and cover the object entirely, a magnetic field in the opposite direction was applied to grasp the object precisely. By applying an RF field at the tip region, F-tips can be merged to lock the object (Fig. 6d-iii). After several seconds of cooling, the

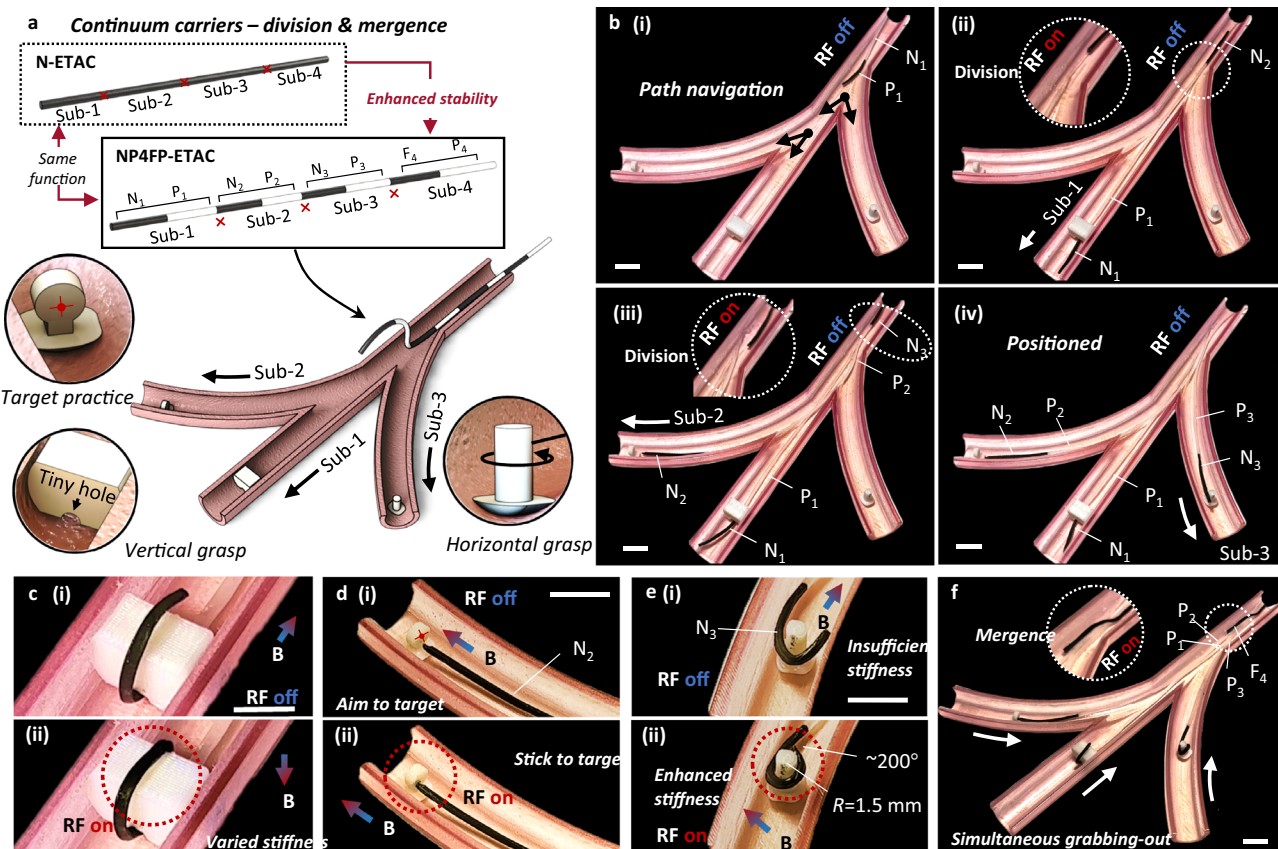

**Fig. 5 | Continuums carriers: division and mergence. a** Illustration of the experimental setup. An ETAC (electromagnetically transitional and actuatable material-based continuum) was designed with a structure of $N_1$-$P_1$-$N_2$-$P_2$-$N_3$-$P_3$-$F_4$-$P_4$. An ETAC can be inserted into the main channel and be divided into three N-P and one F-P sub-continuums. The N-P continuums can independently conduct different tasks, i.e., vertical grasp, target practice, and horizontal grasp. After separately completing their tasks, the N-P continuums can be merged to the F-P continuum at the insertion end, realizing the simultaneous grabbing-out purpose. **b** Self-divisions and separate insertions to subchannels. (i) Path navigation guided by the $N_1$ tip. (ii) Self-division at the $P_1$-$N_2$ connection to release the $N_1$-$P_1$ sub-continuum. The released $N_1$-$P_1$ sub-continuum was further guided to the middle channel. (iii) Self-division at the $P_2$-$N_3$ connection to release the $N_2$-$P_2$ sub-continuum for the top subchannel navigation. (iv) The rest of the continuum was inserted into the bottom channel guided by the $N_3$ tip. Scale bar: 1 cm. **c** ETACs can vary their stiffness to further enhance their flexibility, which is useful for adapting and grasping objects with irregular shapes. Scale bar: 1 cm. **d** The $N_2$-$P_2$ sub-continuum can accurately aim at the target by magnetic controls. By contacting the target and transiting to the softening state, the sub-continuum can stick to the predefined target with a body-acceptable temperature (45–50 °C for ~10 s). Scale bar: 1 cm. **e** To circumferentially grab the rotary body, the $N_3$ tip is required to bend locally at an angle of more than 200°. By applying an RF (radiofrequency) field at the $N_3$ region, its stiffness can be further enhanced and complete the task. The stiffness of the continuum was reduced by more than 1000%-2000% after being remotely softened. Scale bar: 1 cm. **f** The $P_3$-$F_4$ connection can be divided to release the $F_4$-$P_4$ sub-continuum. The $F_4$ tip was guided to the insertion ends of the other three sub-continuums and conducted self-mergences. Therefore, these sub-continuums were assembled back to a single ETAC. By dragging the insertion end of the merged continuum, these objects can be grabbed out simultaneously. Scale bar: 1 cm.

insertion end was dragged to pull out the object (Fig. 6d-iv). Satisfactory flexibility in object captures (i.e., adapting to regular and irregular geometries of different objects) and minimal object disturbances (i.e., no escaping was observed) were demonstrated in the process (Supplementary Movie S11).

To demonstrate the adaptability to objects with different properties, the end-effector successfully removed a paper strip, a piece of animal tissue, and a wooden stick. The paper strip represents a class of objects with high surface energies (~50 mN/m). Taking advantage of ETAMs' adhesive nature, the end-effector can firmly stick to objects' surfaces for removal (Fig. 6e). However, for objects with low surface energies, such as animal tissues (<20 mN/m), the above-mentioned delicate grasping is a suitable strategy (Fig. 6f). Regarding sharp objects, such as pins and wooden sticks, the end-effector can be simplified into a single N-stem to precisely stick to the object for removal (Fig. 6g).

Regrafting endows ETACs with the ability to form new structures in a relatively large space after navigating through bottleneck terrains. With the alignment strategy, the graftable end-effector can be assembled in situ through regrafting. As shown in Fig. 6i, an ETAC was inserted through the glottis and navigated to the trachea under magnetic fields. The ETAC could accurately respond to the magnetic field for bending and conduct self-division by RF heating. By applying a magnetic field in the opposite direction of the separated ETAC's magnetization profile, the separated ETAC could align well with the second arm. Repeating the above steps two more times resulted in four arms aligning with each other, which can merge together to the P-region by activating the RF field. After cooling the heated region, the graftable end-effector (i.e., gripper) was structured and could be utilized to grasp and lock onto the foreign body for removal. See more details in Supplementary Movie S18.

Throughout the entire process, tip-locking is necessary to provide sufficient force for removal. Object escape was observed without the tip-locking function (Fig. 6h). It should be noted that the demonstration was conducted at the joint region of the trachea and bronchi, allowing an end-effector with a relatively large size to be inserted (the trachea outer diameter, OD, in this case, is ~2 cm). According to the previous section of this work, the ETACs can be further scaled down to

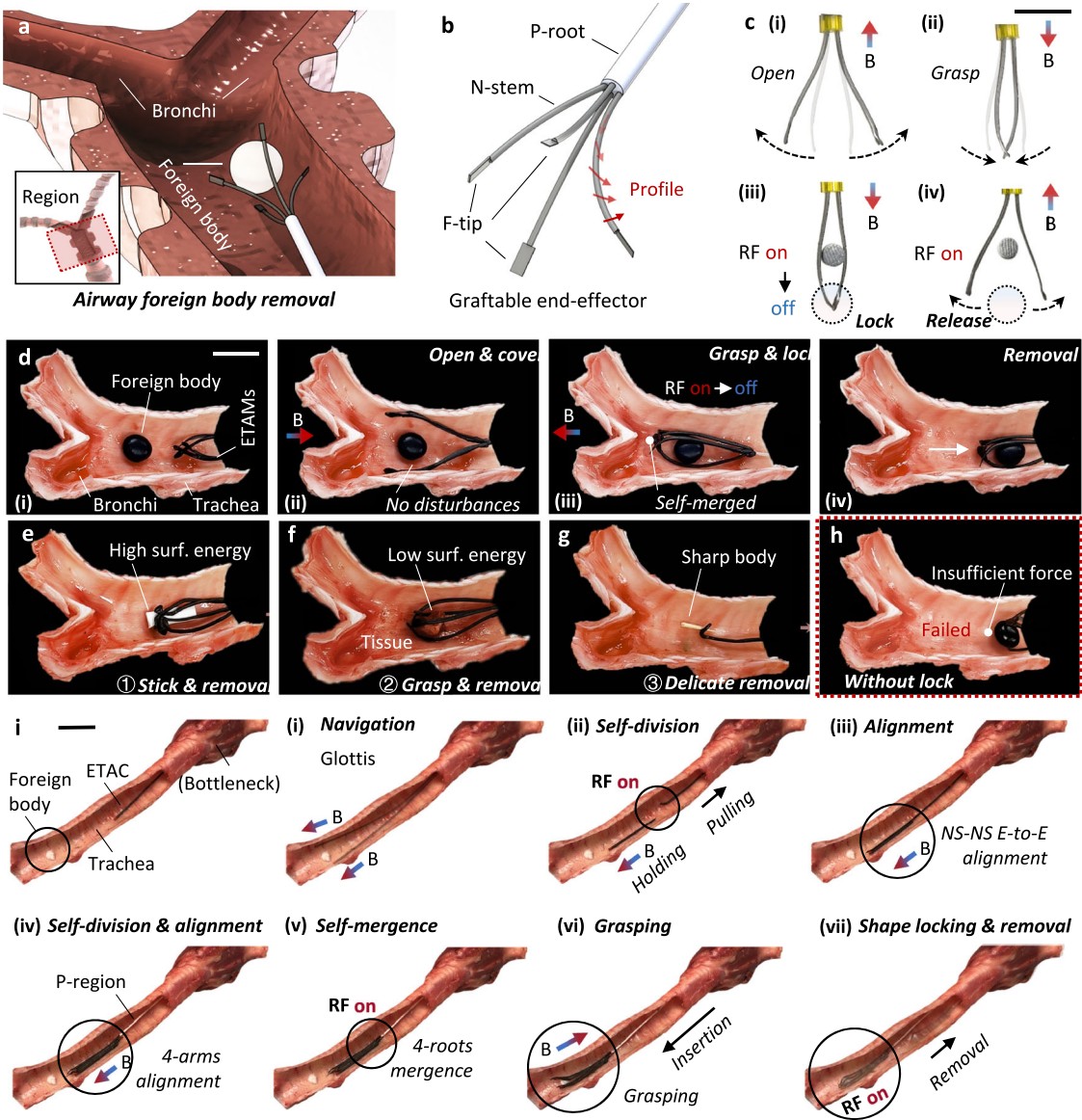

**Fig. 6 | Graftable end-effector: Airway foreign body removal. a** Illustration of the foreign body removals at the airway. **b** The design principle of the graftable end-effector. The end-effector consists of a P-root and four N-stems with F-tips. Red arrows denote the magnetization profile of N-stems. **c** The end-effector can respond to external magnetic fields, performing opening and grasping motions. To provide sufficient grasping forces, the F-tips of the end-effector can bond to each other by self-mergences and unlock by self-divisions. Scale bar: 1 cm. **d** Foreign body removal process at an ex vivo pig bronchi system. (i) The end-effector was inserted into the trachea and approached the foreign body. (ii) Responding to an external magnetic field, the end effector opened and covered the object without any disturbances. (iii) The end-effector grasped the foreign body and locked its F-tips to firmly constrain the object. (iv) The object was stably removed by pulling the P-root. Scale bar: 1 cm. **e**–**g** Objects with different properties were successfully removed. ① Objects with high surface energy can be removed by the bonding strategy. ② An animal tissue with low surface energy can be removed by the grasping strategy. ③ A sharp body can be removed by the bonding strategy of a single N-stem. Scale bar: 1 cm. **h** A case showing the necessity of the tip locking. Without the locking, tips were observed with insufficient force to remove the object. Testing results showed that the end-effector with a lock can gain ~400% strength enhancement than that without a lock. **i** In situ gripper assembling by regrafting and alignment strategy. The processes include (i) navigating through the glottis to the trachea under magnetic guidance, (ii-v) multiple times regrafting and alignment to structure an end-effector, (vi) grasping under magnetic actuation, and (vii) shape locking for removal.

---

fulfill the smaller size requirement for operations in the bronchial system (e.g., OD < 1 cm). Regarding the operating temperature consideration, the phase change process was conducted within a range acceptable for the human body (45–50 °C for ~10 s) and without interaction with tissue surfaces. We also note that the grasper can be assembled in situ, as shown in Supplementary Materials S13.

Endoscopic submucosal dissection (ESD) is a minimally invasive procedure to remove precancerous and cancerous areas in the GI tract[29,30]. Our previous experience in designing ESD tool systems and operating ESD procedures showed that complicated rigid mechanical tools need to be further simplified and integrated[10]. For example, when

cancerous tissue without a surgical biopsy requirement needs to be dissected, the procedure steps can be simplified into three main steps: thermal ablation, pulling assistance, and dissection. In this regard, involving ETACs holds significant potential in procedure simplifications.

We designed an ETAMs-aided surgical device group, containing ETACs and an ETAMs-aided electrosurgical unit (ESU, Fig. 7). The ETAC was designed with a structure of P-F regions, which were applied to conduct thermal ablations and pulling assistance.

The ETAMs-aided ESU, covering an ESU by N-ETAMs, can be actuated both manually and magnetically for posture adjustment

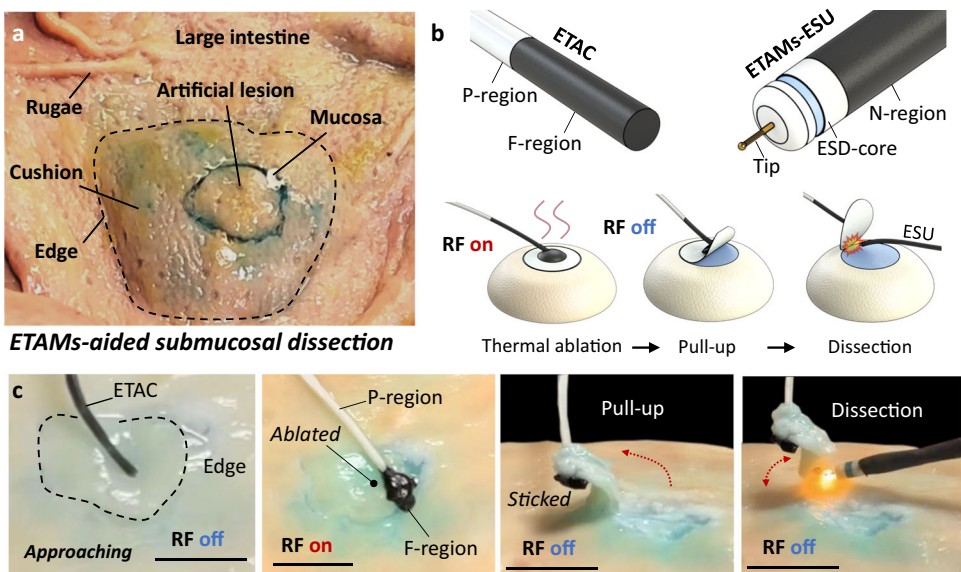

**Fig. 7 | Aided submucosal dissection. a** Experimental setup for submucosal dissections at an ex vivo pig intestine. **b** An ETAMs (electromagnetically transitional and actuatable material-based materials)-aided surgical device group contains an ETAC with a structure of P-F regions and an ETAMs-aided electrosurgical unit (ESU). The surgery process includes thermal ablations, pulling assistances, and collaborations for dissections. (**c**) Real-time images demonstrating the submucosal dissections operated by the ETAMs-aided device group. Scale bar: 1 cm.

(Fig. 7b). To showcase the entire submucosal dissection process, we built an experimental setup on the inner wall of an ex vivo pig intestine (Fig. 7a). The ETAC was first inserted to approach the working region, contacting the artificial lesion by its F-tip. Applying an RF field in 3–4 s transformed the F-tip into fluids with a high temperature (~120 °C) to thermally ablate the target tissue. This process can vaporize the mucous for tissue adhesions while ablating the tissue. After turning off the RF field, the melted ETAMs returned to the elastomer state and bonded to the ablated tissue. By pulling the insertion end of the continuum, the tissue could be partially lifted, exposing the under-dissected regions of the artificial lesion. Consequently, the ETAM-aided ESU can be inserted to approach the target region, conducting cutting to remove the tissue (Fig. 7c, Supplementary Movie S12). The demonstration primarily proves ETACs' potential for operating complicated tasks. Several technical issues need to be solved in future studies. For example, considering heat diffusion, the ablation effects of ETACs should be further investigated to guarantee that surrounding tissues will not be harmed. We also proposed an advanced function called in vivo meshing, which follows a similar concept to in vivo printing technology to complete ESD by multiple point ablations and multiple region ablations (please find the details in Supplementary Materials 24 and Supplementary Movie S17).

To further investigate the heating effect of thermal ablation, we conducted GI tract ablation both ex vivo (on porcine stomach) and in vivo (on nude mouse esophagus and stomach). Regarding the ex vivo porcine stomach ablation, four regions were preset for comparisons (Fig. 8a): region-1 as the control group, region-2 as the 10 s heating group, region-3 as the 30 s heating group, and region-4 as the 60 s heating group. H&E staining was conducted on these tissues, and the results showed that different degrees of inflammatory cell infiltration at the ablation site (stomach inner wall). The results revealed that the heating effect increases with the increased heating time (Fig. 8b).

Regarding the in vivo mouse test, an ETAC branch (containing an endoscopic ETAC for imaging guidance and F-ETAC for ablation) was applied to conduct RF ablation (RFA) in the GI tract (Fig. 8c). The upper esophagus (OD < 2 mm), cardia, and stomach were chosen as the ablation regions (Fig. 8d). The whole surgery process includes the insertion, in vivo ablation, extraction, dissection, and ex vivo re-

validation. Under external magnetic guidance, the ETAC was inserted from the oral cavity and navigated through the esophagus. RF was turned on (for ~30 s) after the ETAC accurately arrived at the preset target regions. After thermal heating, the ETAC was extracted, and the ablated tissues were dissected and H&E stained for observations. Ex vivo stomach ablation was also conducted on the dissected mouse stomach for validation. Results show that ablation processes effectively kill cells in contact and damage the tissue structures (Fig. 8e). We also note that a small number of F-ETAM particles were left at the ablation region, which is acceptable due to the satisfactory biocompatibility of ETAMs (Supplementary Materials–Section 6, biocompatibility test).

Regarding heat generation in vivo, the existing literature has provided safety boundaries on temperature ranges (as shown in Fig. 8f). Divided by the epidermal injury temperature curve and complete irreversible damage temperature curve, the heating temperature-time area contains the body temperature range, moderate ablation range, and ideal therapeutic ablation range. The body temperature range refers to the commonly acceptable working temperatures of biomedical devices. The moderate ablation range is applied for liver, pleural, and abdominal tumor RFAs. The ideal therapeutic ablation range was proven to be a satisfactory operation range between immediate coagulation and tissue vaporization/carbonization, which has relatively high surgery efficiency. In this work, the ablation temperature (~120 °C) falls within the ideal therapeutic ablation range, which aligns with existing studies.

Given the regrafting function, ETACs hold the potential to conduct procedures requiring passing through narrow bottlenecks (Fig. 9), such as endoscopic retrograde cholangiopancreatography and transurethral procedures, which the common functional continuum cannot easily insert. ETACs also have the potential to operate minimally invasive surgeries for small-size animals, such as nasal passage intubation, RF ablation in the gastrointestinal tract (demonstrated in Fig. 8), and respiratory system navigation. The abovementioned lumens are of narrow sizes ranging from 1 mm to 5 mm, which are challenging for existing continuum robots to navigate and operate complex tasks. We have also added a list showing ETAC's potential biomedical application scenarios in Supplementary Materials S27.

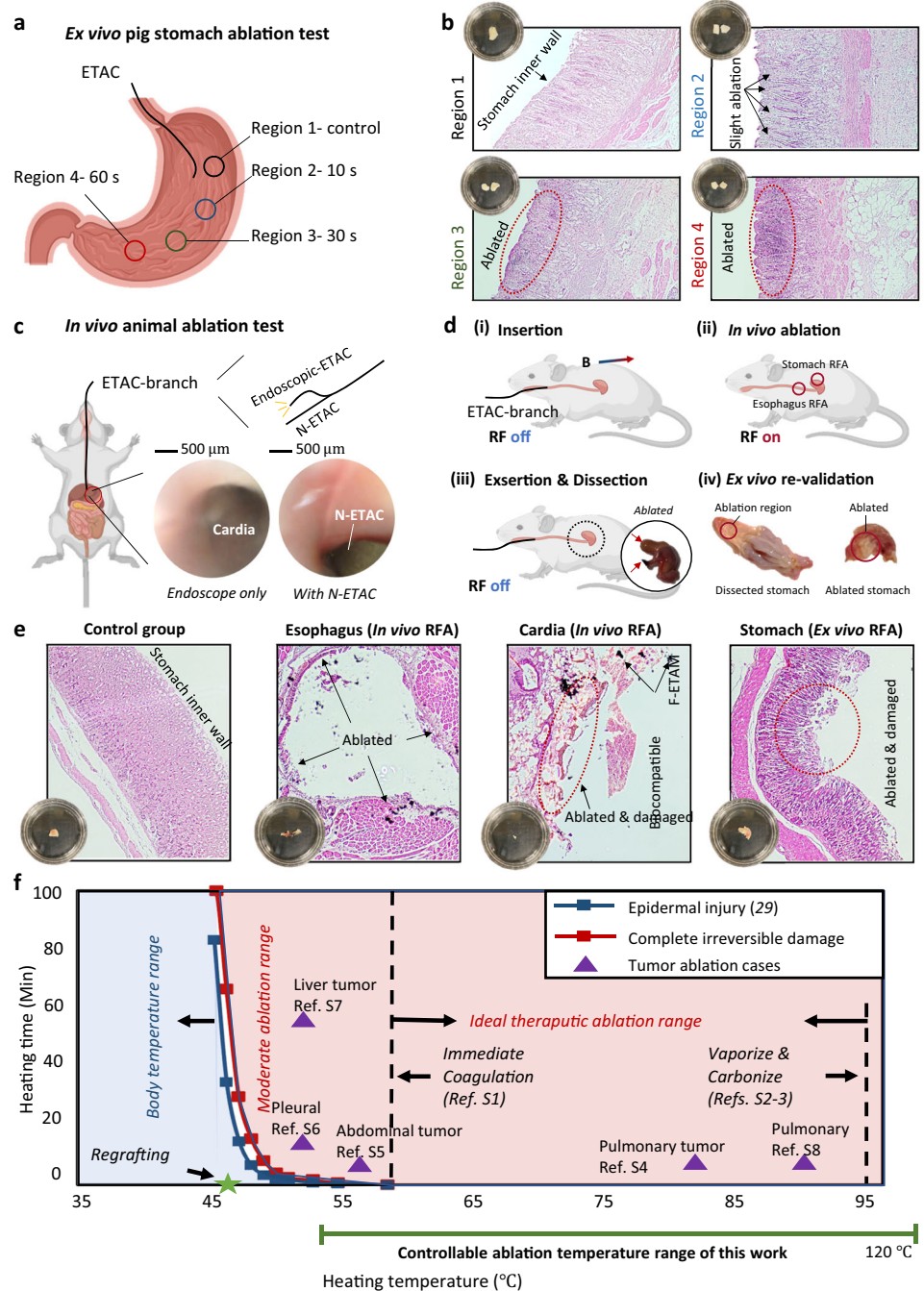

**Fig. 8 | Ablation test and analysis. a** Ex vivo ablation tests were conducted on porcine stomachs. Four regions were selected as the control group: 10 s ablation group, 30 s ablation group, and 60 s ablation group. **b** Hematoxylin-eosin staining was conducted on the tested tissues, showing that the area of the inflammatory cell infiltration region increases with the increment of heating time. **c** An in vivo ablation test was conducted on nude mice, where an ETAC branch was applied for imaging and ablation. **d** The whole surgery process contains insertion, in vivo ablation, and exsertion. After the surgery, the ablated regions were dissected and H&E stained to validate the heating effect. **e** All tested tissues were H&E stained and observed, including a stomach control group, ablated upper esophagus tissue, ablated cardiac tissue, and ablated stomach tissue. **f** A comparison study between existing ablation cases and this work. Source data are provided as a Source Data file. Created in BioRender. Yang (2025) https://BioRender.com/f9oc6i9.

Within an actuation distance that is more than 10 cm, the proposed ETACs could perform their characteristics in trachea-bronchi navigation, endoscopic retrograde cholangiopancreatography, transurethral procedures, endoscopic submucosal dissection, and small-size animal nasal passage intubation, RF thermal ablation in the gastrointestinal tract, respiratory system navigation.

## Discussion

We proposed submillimeter/millimeter-scale regraftable FSCs called ETACs. While exhibiting comparable bending performances with existing devices, ETACs were further endowed with regraftability, demonstrating the functionality of completing complicated tasks in confined spaces for biomedical purposes, such as endoscopic grasping

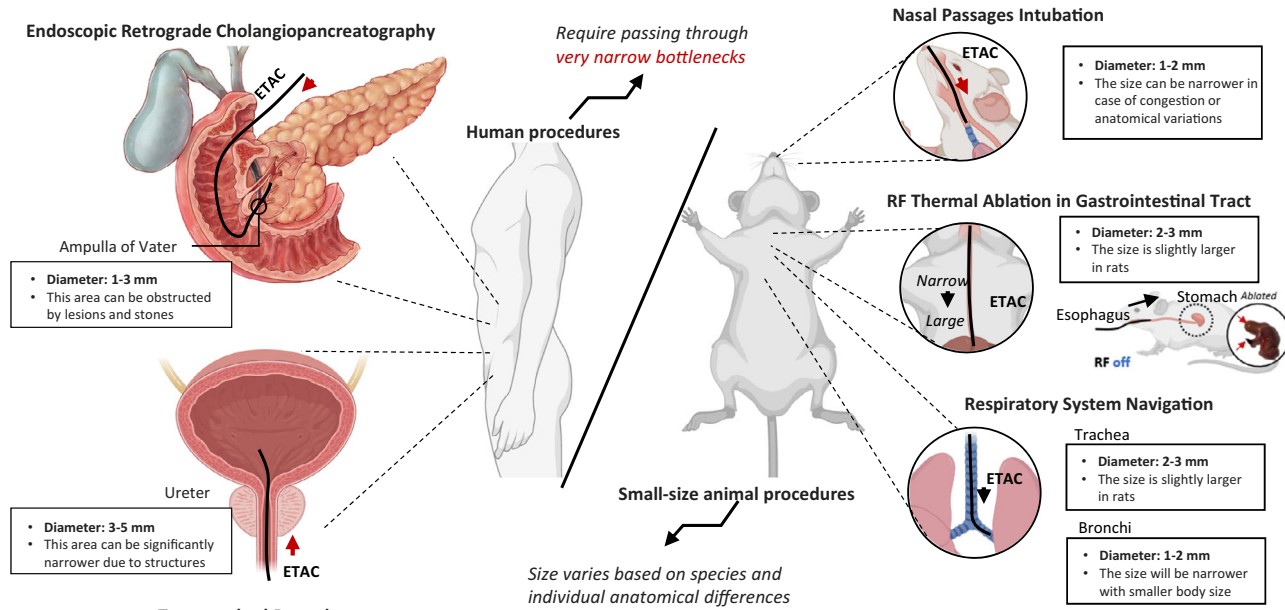

**Fig. 9 | Biomedical application scopes of ETACs (electromagnetically transitional and actuatable material-based continuums) requiring passing through narrow bottlenecks.** Human procedures include endoscopic retrograde cholangiopancreatography and transurethral procedures. Small-size animal procedures include nasal passage intubation, RF (radiofrequency) thermal ablation in the gastrointestinal tract, and respiratory system navigation. Created in BioRender. Yang, Y. (2025) https://BioRender.com/pbrbpao.

and thermal ablation. To prove the above, we conducted experimental studies in the fields of material science, fabrication technology, large deformation analysis, thermodynamic investigation, magnetics, and biomedical engineering.

ETAMs' elastomer-fluid phase change nature was first explained by microscopic analysis. Because of the thermoplastics-based material, ETAMs are suitable for the fused extrusion printing strategy. Therefore, we designed a corresponding printing platform and manual printing method to showcase the fabrication practicality of ETACs.

We endowed ETACs with the steerability capable of performing challenging bending motions that current continuums cannot achieve. We conducted experimental and numerical studies on the basic bending function of ETACs. Navigation was accomplished in a complex aerodigestive tract, a bronchi phantom, and an ex vivo pig bronchial system.

The next consideration is *how* and *why* FSCs should be endowed with regrafting. Regarding in vivo manipulation and therapy, existing submillimeter-scale FSCs are hindered by the tiny sizes and simple structures. Existing continuums with coated layers face difficulties in realizing in situ tetherless divisions and mergences, which are necessary for advanced manipulations, such as simultaneous grasping (Fig. 5), self-locking/releasing (Fig. 6), and melting ablation (Fig. 7) etc. For example, an LMPA-enabled continuum undergoes sensitive phase changes between solid and fluid states, thereby needing layer coatings for protection, which makes regrafting challenging. Moreover, ETACs can perform elasticity-plasticity transitions for steering enhancement (from ~100° to ~200°) and desirable softenability (1000%-2000% stiffness softening effect), which is difficult for current continuum materials to achieve. Due to the suitable properties of ETAMs (i.e., elasticity, softenability, and fluidity), regraftings were successfully realized in ETACs. Through multi-physical analysis of magnetism and thermodynamics, self-division and self-mergence were respectively proven to conduct divisions by multiple strategies and to operate magnetization re-programmability for various purposes.

With the fundamental mechanism understanding of self-divisions and self-mergences, ETACs were further applied to three biomedical

scenarios to showcase the potential enhancement of current surgical technologies. Before discussing the application details, it is important to explain heat generation during operations. There are three heat generation-related terms: preheating temperature, softening temperature, and ablation temperature, which should be distinguished for a better understanding (see detailed heat safety analysis in Supplementary Materials—Section 10).

Although all division strategies were successfully realized, we found that self-division is more complicated to conduct than self-emergence. There are two reasons: (1) compared with the self-emergence that commonly requires a single RF heating field, the self-division requires both heating and external magnetic moment for splitting. (2) The magnetic moment generated by the field is required to be relatively large for division. Moreover, a stable rotational magnetic field with an appropriate rotation speed is required in specific cases, such as twisting division. Regarding the large bending back-and-forth in Fig. 3g and its corresponding movie, such a phenomenon was also avoided in the abovementioned demonstration. The large bending back-and-forth was led by the insufficient heating temperature and the relatively low magnetic strength of the utilized small permanent magnet. By increasing the heating power and utilizing a bigger permanent magnet, the large bending back-and-force was avoided, as shown in Fig. S80 and Supplementary Movie S20.

The preheating temperature is applied to softened ETACs ex vivo to 60 °C for 10 min, which leads to enhanced steering ranges while maintaining the softening effects for 9–10 h.

The softening temperature (i.e., 45–50 °C for ~10 s with/without direct tissue contacts) is applied for in vivo regrafting. In the "continuums carriers" (Fig. 5) and "regraftable end-effector" (Fig. 6) demonstrations, the softening temperature represents the highest temperature generated. It is important to mention that this temperature range is considered safe for human body applications, given the contactless and short-duration local heating methods employed[31].

The ablation temperature (i.e., 120 °C for 10–30 s) is applied for thermal ablations in the submucosal dissection demonstration (Fig. 7).

It is important to note that such high temperatures are generated only for thermal ablation purposes and were not used in any other demonstrations provided.

The "continuums carrier" (Fig. 5) demo showed in situ divisions and mergences for simultaneously grabbing multiple objects. ETACs were found to perform shape memory, vary stiffness, and bond targets during the process. In the airway foreign body removal demo, ETACs were fabricated into a graftable end-effector that can grab objects with different properties (i.e., geometries and surface energies). This is achieved by tip-locking and unlocking enabled by self-mergences and self-divisions, respectively. Minimal disturbances to foreign bodies were demonstrated during the grasping process (i.e., no object-escaping behaviors were observed), proving that ETACs can enhance the success rate of foreign body removal surgeries.

Moreover, the submucosal dissection demo showcased that ETACs could aid existing surgeries, improving surgical efficiency of tissue retraction and lesion exposure (e.g., tissue retracts in 5–10 s in our case) and enhancing the steerability and manipulation of existing medical devices (e.g., ETAM-ESU). The ETACs were further applied to ablate ex vivo porcine stomach and in vivo mouse GI tract, showing the feasibility in completing challenging biomedical tasks. It should be noted that ETACs are developed for air-laden passages, and the fluid-laden lumens are outside the scope of this work due to the completely different procedure requirements, such as heat generation/dissipation and environmental flow velocities.

Regarding the pulling force in the foreign body removal case, we conducted a series of pulling tests using a 0.1 mm × 6 cm continuum. The results indicate that the continuum made of N-ETAM can lift objects with a weight of 600 g (6 N), while the continuum made of P-regions can lift 700 g (7 N). Theoretically, an end-effector can generate a maximum pulling force of 24 N, which is acceptable for foreign body removal[32]. It reveals that regraftable continuums are sufficiently stiff to withstand external loads and have a low risk of leaving segments inside the human body.

Regarding adhesion to the wet tissue surfaces during thermal ablation, the fluidified ETAM vaporizes the liquids (i.e., mucous) and directly bonds to the tissue surface, thereby providing sufficient adhesion forces. During the experiments, no debonding phenomenon was observed between the ETAM and ablated tissues during the lifting process. ETACs can also conducting in vivo meshing for more complicated operations which is shown in Supplementary Material S24.

The biocompatibility of biomedical materials should be considered. According to previous studies, PCL and magnetite particles were well-proven biocompatible in predefined particle dimension ranges[33,34]. However, the solutions of neodymium MNPs are still under discussion due to their non-neglectable toxicity[35]. In this work, we co-cultured HeLa cells with ETAM samples (i.e., N-ETAM and F-ETAM). Cell counting kit-8 (CCK-8) assay results showed that the viability of cells co-cultured with N-ETAM and F-ETAM for 72 h was 79.35% and 86.50%, respectively. Moreover, the median fluorescence intensity of the co-cultured cells showed that the live/dead cell ratio for the N-ETAM group was slightly lower than the F-ETAM group (i.e., 0.9744 vs 0.9831). According to the above-mentioned preliminary biocompatibility test, we conclude that both N-ETAMs and F-ETAMs have the potential for biomedical applications (see details in Supplementary Material Section 6). However, further biocompatibility tests should be conducted for practical applications. Note that the silica coating of MNPs is also applicable to ETAMs, which were preliminarily proven effective for FSCs[1].

Overall, the proposed regraftable FSCs, ETACs, may pave the way for endoscopic continuums to operate with more advanced flexibility and functionality, thereby accomplishing complicated tasks in confined spaces, especially in biomedical scenarios.

## Methods

The mice experiement is approved under the government of the Hong Kong Special Adminstrative Region Department of Health: Licence to Conduct Experiments, including HR (24-423) in DH/HT&A/8/2/1 Pt.63, YY (24-424) in DH/HT&A/8/2/1 Pt.63, and WS (24-365)inDHHT&A/8/2/1 Pt.62. The animal surgery protocal is granted by Animal Experimentation Ethics Approval of The Chinese University of Hong Kong under Ref. No. 24-023-GRF.

### ETAMs preparation

PCL particles (150 μm, Ruixiang Plastics Inc.), neodymium MNPs (5 μm, Magnequench Co., Ltd.), and magnetite NPs (48 μm, Leber Inc.) were selectively weighted and premixed in a glass culture dish, which was heated by a heating platform (Yarun Inc.). Heating temperatures varied from 160 to 200 °C with respect to different PCL particle types. Regarding the thermal mixing, the glass culture dish was preheated for ~5 min on the heating platform, after which all weighted particles were added and stirred for another 5 min. At this time, microparticles can be dispersed well in the melted PCL by mixing at a speed of 120 r/min for 5 min. By keeping the same stirring speed, the temperature is then lowered to the temperature of PCL softened state. After 5 min of stirring, the mixture can be quickly cooled to ambient temperature by placing the culture dish into cold water. It should be noted that the first round of mixing takes advantage of PCL's low viscosity under high temperatures, while the second round ensures the even dispersion of the microparticles due to the moderate viscosity of ETAM. In Supplementary Movie S1, phosphor powders were mixed into the ETAMs to distinguish the P-regions and N-ETAMs. In practical cases, phosphor powders were not involved.

However, no obvious performance differences were observed when comparing ETACs fabricated by the above steps to those fabricated with a single stirring process under high temperature. This result indicates that the viscosity range of the melted PCL allows for effective dispersion while preventing the particles from settling.

### Microscopic observations

An optical microscope (BX60, Olympus Inc.) and an SEM (Q400F field emission SEM, FEI Inc.) were applied to conduct the microscopic observations. A silicone rubber mold (EcoFlex 0010, Smooth-on Inc.) was fabricated to prepare ETAMs samples (1 × 1 mm cubes) for observation. In addition to visually observing the ETAM sample, element surface scanning analysis was conducted to investigate the element distribution on ETAMs' surfaces (see details in Supplementary Materials Section 1 and Section 2).

### Bending tests

ETACs with homogenous stiffness tips ($L_M/C$ studies) were tested in a Helmholtz coil (Paisheng tech. Inc.). Magnetic field strengths of 4, 8, 12, 16, and 20 mT were selected to actuate the continuums. Regarding the varied stiffness study, continuums with a total length of 3 cm were actuated by a permanent magnet (48 × 48 × 48 mm cube, Haoci Magnets, Inc.) for a stronger gradient field.

### Numerical simulations

A magnetics-hyperelastomer coupled numerical model was developed with COMSOL Multiphysics (COMSOL Inc.) to simulate the bending behaviors of the graded stiffness tips. The Neo-Hookean hyperelastomer model was applied to the continuums, and the residual magnetic flux density model was selected to describe the magnetization profiles of the continuums. Maxwell tensors at the tip surface were chosen as the interface between the magnetics solver and hyperelastomer solver. Uniform magnetic fields were applied to describe the EM actuation process. For PM cases, a gradient field was applied instead and the Maxwell tensors at the tip surface were replaced by an

entire body response definition (see details in Supplementary Material Section 5).

## Induction heating

An induction heater (Honghe Inc.) was used to provide RF fields for related studies in this work. With a frequency of 770 kHz, the induction heater provided efficient heat transformation to ETAMs. A solenoid-shaped and a disk-shaped coil were installed on the heater to generate different RF fields. In Fig. 3b, two plastic holders were printed (J826, Stratasys Inc.) to control the heating distance quantitatively.

## Demagnetization test

ETACs were magnetized by a capacitive pulse magnetizer (Yuanxing Inc.). The N-P-F-P-N continuum was pre-magnetized in the axial direction and put into a silicone rubber mold for testing. Twelve probe points were given to the 90 mm continuums, measured by a magnet-ometer (Tianheng Measurement Inc.). After installing the continuum, the mold was heated by a temperature-controllable heating platform for 10 min. The magnetization moments were recorded before and after the heating.

## Stretching test

The No.2-typed (2#) ETAMs specimens were prepared for stretching test by a mechanical testing machine (Instron Inc.). The testing process followed the protocol of ISO/DIS37-1990. The test region was dimensioned with length×width×thickness of $20 \times 4 \times 3$ mm$^3$. The obtained strain-stress data was nonlinearly fitted by the Neo-Hookean model. Note that the linear region of the obtained data was selected because the plastic deformation was not observed during bending tests.

## Movement precision of ETAM-ESU

Regarding the movement precision of the ETAM aiding the ESU, we have conducted a series of movement precision tests. In the continuum control area, there are two main test methods to determine the moving precision of a continuum robot: movement resolution and position repeatability. As shown in Fig. S63a, we installed an ETAM-ESU in a Helmholtz coil where a grid mesh was placed beneath the ESU for motion measurement. The magnetization profile of the ETAM-ESU was preprogrammed along the axial direction and the magnetic field direction was perpendicular (defined as y-direction) to the magnetization profile for bending actuation. The movement resolution results are shown in Fig. S63a-c. The ETAM-ESU performs satisfactory linear controllability with the increment of actuation voltage (namely magnetic moment). As shown in Fig. S63d, the repeatability showed ~0.3° averaged orientation error and ~0.6 mm averaged positioning error. It was observed that the position repeatability is higher in the low magnetic moment range than in the high moment range. This is because the positioning error accumulates with the increment of the magnetic moment.

## Biocompatibility

N-ETAMs and F-ETAMs samples were prepared into disks with a diameter of 3 mm and a thickness of 0.6 mm. HeLa cells were co-cultured with ETAMs samples in a 96-well plate. Five biological replicates were used for each group. Material sterilization was conducted before using the CCK-8 assay to investigate the cell viability. The samples were first soaked in medical-grade alcohol for 30 min and rinsed 3–5 times with sterile phosphate buffer saline solutions. Ultraviolet germicidal irradiation was finally applied to the samples before co-culturing them with HeLa cells for 24, 48, and 72 h. Cell viability and mean fluorescence intensity (MFI) of Calcein acetoxymethyl ester (Calcein-AM, for live cells) and Propidium Iodide (PI, for dead cells) were calculated for the co-cultured cells (see detailed results in Supplementary Material– Section 6).

## Animal organ preparations

The ex vivo pig bronchi system was peeled from a commercially available pig lung. Trachea (inner diameter of ~20 mm), main bronchi (inner diameter of ~9 mm), superior lobar bronchi (inner diameter of ~6 mm), middle lobar bronchi (inner diameter of ~4 mm), and inferior bronchi (inner diameter of ~2 mm) were pealed. The pig large intestine used for the submucosal dissection was commercially obtained. The large intestine was pre-dissected for artificial lesion setup. 2–3 mL agent (a mixture of normal saline and methylene blue) was injected manually into the submucosal layer to form a cushion (~3 cm) at the injection zone (see details in Supplementary Material– Section 7).

## Animal trial and post-processing

The protocol of endoscopic ablation in the rats and mice was approved by the Laboratory Animal Services Centre at The Chinese University of Hong Kong.

A nude mouse (27.428 g) was selected for the in vivo ablation procedure. The mouse was fasted for 12 h to ensure an empty stomach for optimal continuum insertion. Anesthesia was induced in the mouse by injecting 0.5 ml Ketamine, 0.25 ml Xylazine, and 4.25 ml phosphate buffered saline (PBS) before the procedure.

The mouse was positioned within a solenoid-shaped induction coil, and the ETAC branch was introduced through the oral cavity into the esophagus. Once the continuum accurately reached the predetermined targets (upper esophagus, cardia, and stomach), the RF field (770 kHz) was activated for 30 s. The ETAC branch was removed after approximately 1 min of cooling down.

Following the procedure, the mouse was euthanized by overdosing ketamine (100 mg/kg) and xylazine (16 mg/kg) before dissection. The esophagus and stomach tissues were excised and preserved in a 10% Formalin solution.

Samples were fixed in 10% formaldehyde for 24 h, dehydrated using gradient ethanol, vitrified by xylene, and embedded in paraffin. 4 μm thick sections were prepared with a rotary microtome (NM 325; Leica Microsystems, Germany). The section of tissues was deparaffinized by xylene and ethanol and rehydrated. To evaluate the tissue morphology, the slides were stained with Harris hematoxylin solution for nuclei and then washed in the following sequence: running tap water, acid alcohol, tap water, Scott's buffer, and running tap water. Furthermore, the samples were stained with 1% Eosin Yellowish solution. Finally, the slides were gradually dehydrated by ethanol and xylene and mounted with a coverslip.

## Heating efficiency

We regard this issue as an industrial problem that will be effectively addressed by increasing the size and power of the RF machine[36]. Here, we provide our rationale regarding the concerns on heating efficiency and resulting field safety in detail. Current laboratory RF equipment is limited by its size and power, which are much lower than existing industrial and medical devices[36,37]. For example, the RF frequency of existing medical devices ranges from 128 MHz (3 T) to 342 MHz (8 T), which is more than 400 times higher than our laboratory RF machine. While providing satisfactory RF efficiency, such industrial RF machines can support relatively long-distance heating. The maximum coil diameter we found is 56 cm, which is acceptable for practical application[36,37]. Regarding the field safety considerations, there is research on animal safety evaluation for extreme RF exposure. Researchers found that 10–12 T RF exposure for a continuous 28 days was relatively safe for mice. It was also reported that 3.5–23.0 T RF exposure for 2 h and 7.0–33.0 T for 1 h did not have severe long-term detrimental effects on mice[38,39]. Therefore, we regard the heating efficiency and field safety as well-proven issues, which can support our proposed regrafting technique.

## Reporting summary

Further information on research design is available in the Nature Portfolio Reporting Summary linked to this article.

## Data availability

All data are provided in the manuscript and the Supplementary Materials. Source data are provided with this paper.

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

## Acknowledgements

The authors thank Dr. Huxin Gao for the discussions and help in setting up the submucosal dissection demonstration. This work was supported by Hong Kong Research Grants Council (RGC) Collaborative Research Fund (CRF C4026-21G), Research Impact Fund (RIF R1007-24), General Research Fund GRF 14204524, 14203323, 14216022; Research Grants Council (RGC)-NSFC/RGC Joint Research Scheme N_CUHK420/22; National Science Fund for Young Scholars, and Shenzhen-Hong Kong-Macau Technology Research Programme (Type C) STIC Grant 202108233000303.

## Author contributions

H.R. conceived the original idea, designed the experiments, supervised this work, provided critical revisions, and contributed to the conceptual design and manuscript editing. Y.Y. conceived the original idea, designed the experiments, drafted the manuscript, and was responsible for its revision. W.S. participated in the experiments and assisted with manuscript editing. T.X. performed the biocompatibility experiments. B.Y. carried out the tissue section experiments. Z.L. designed the biocompatibility and tissue section experiments and participated in manuscript editing.

## Competing interests

The authors declare no competing interests.
