## [Transparent Peer Review file · Nature Communications]

Regrafting Submillimeter-scale Ferromagnetic Soft Continuums

Corresponding Author: Professor Hongliang Ren

Version 0:

Reviewer comments:

Reviewer #1

(Remarks to the Author)

The manuscript presents a novel design of FSCs that utilizes self-graftable capabilities—combining self-division and self-mergence—to navigate and function in confined and tortuous endoluminal scenarios. This is achieved through the development of ferromagnetic thermoplastic soft materials, which allow for reversible elastomer-fluid transitions, facilitating arbitrary division-mergence actions. The authors systematically explore the mechanisms underlying these functionalities and demonstrate the practical applications of this technology in performing complex medical tasks like targeted delivery, foreign body removal, and thermal ablation in simulated environments. Regarding the experimental demonstration, robot design, self-division, and self-mergence, the reviewer should pose the following concerns.

Major issues:

Comment #1 :

It is important to note that the dimensions of continuum robots are determined by the size of the endoluminal scenarios in which they operate. Robots used in neurointerventional or cardiac interventional surgeries require very small sizes, which results in the medical devices they carry being less sophisticated. In contrast, endoscopies intended for the respiratory, digestive, and urinary systems are not subject to such stringent size requirements. There are many endoscope-compatible medical devices available for these systems, capable of performing tasks such as foreign body retrieval, lithotripsy, electrosurgery, and radiofrequency ablation, all of which have been clinically tested. It is necessary to limit the scope of application of the proposed continuum robots.

Utilizing smaller robots can reduce invasiveness and enhance patient-friendliness, but this naturally presents challenges in functional integration. If it is possible to reduce size while maintaining reliability and functionality, that would represent a significant advancement. The grafting functions proposed by in this work are an effective potential method for functionalizing small-sized robots. However, three experimental demonstrations presented in the manuscript predominantly showcase the application of controllable adhesion, and do not effectively demonstrate the claimed division and merger functions. Beyond connections, division and merger also require controlled movement of both ends. For instance, navigating a robot through a confined lumen to an open space, where it can self-divide into multiple untethered segments that merger into a functional structure larger than the orifice to perform tasks, is beyond the capability of existing robots and can strongly support the arguments made.

1. For the first demonstration (Movie S10, Continuum Carriers: Division and Mergence): The author demonstrates how by breaking and merging, the robot is endowed with three tips, allowing it to remove foreign bodies from multiple channels simultaneously. The endoluminal environment is sufficiently large to accommodate the functional structure formed by the robot. In such large channels, would it be feasible to manually assemble this structure externally before commencing the task, or could directly inserting three separate continuum robots at the junction to perform tasks independently achieve quicker results?

2. For the second demonstration (Movie S11, Graftable End Effector: Airway Foreign Body Removal"): The robot is formed into a grasper, and effective capture is achieved using the adhesive properties of PCL. Was this grasper assembled in situ or manually assembled externally? What are the advantages over existing bronchoscopic forceps?

3. For the third demonstration (Movie S12, Assisted Submucosal Dissection"): How are the two continuum robots independently operated? Considering the intestinal wall as a working environment in a natural orifice transluminal endoscopic surgery, arranging these two instruments in an appropriate posture at the surgical site is a non-trivial issue. Particularly since both tools are fine, have an elastic modulus only in the MPa range, and are magnetic, significant mutual interference cannot be ignored. What is the movement precision of the ETAM aiding the ESU?

Comment #2 :

In reviewing the design aspects of the robot as presented in the paper, the following concerns are raised:

1. Temperature monitoring of PCL: Since PCL is not encapsulated, maintaining its temperature below the melting point is crucial to ensure the stability of its form. How does the study monitor the temperature of PCL to guarantee the structural stability and safety of the surgical procedure, particularly given the challenges associated with measuring the temperature of robots in body using infrared thermography?

2. Lubrication needs: The demonstrated robot lacks a lubricating coating or lubricant on its surface. However, lubrication is essential during the deployment of continuum robots, not only to facilitate intubation but also to minimize tissue damage (Adv. Material., 2019, 31, 1807101). In clinical settings, lubrication is also an indispensable operation. Could lubrication enhance the performance of the proposed continuum robot, for example, by facilitating the smooth navigation of the robot with a uniform elastic modulus as shown in supplementary Movie S2? Moreover, would the use of lubricating coatings or lubricants impact the self-graftable capabilities of PCL within the ETAC? And considering the in vivo experiments conducted on mice, it is important to assess how the naturally secreted fluids within bodily conduits, which provide lubrication, might affect the self-graftable capabilities of the robot.

3. Material state of ETAM: PCL's elastic modulus decreases with increasing temperature (Soft Robot., 2022, 9, 189-200). The mechanical properties of ETAM should be given specific values to differentiate between the three states (the elastomer, softening, and fluid state). Considering the heat dissipation of liquids in clinical applications, could this lead to changes in the elastic modulus of continuum robots, affecting the numerical control of tip.

Comment #3 :

The clarification regarding the type of magnetic field device employed in Movie S3 is necessary—was it a coil or a permanent magnet? Particularly in the navigation to target 19, there are two locations prone to buckling. Wrapping a helix around the continuum robot can prevent buckling by transferring thrust from the rear to the tip (Sci. Robot., 2024, 9, eadh0298), how was buckling avoided in this study?

Comment #4 :

About the self-division capability of the proposed system, the following concerns are raised:

1. Heating efficiency of ETAM: The heating efficiency of ETAM decreases sharply as the distance from the coil increases. As shown in Fig. 3C, a distance of 25 mm significantly reduces heating efficiency, which is much less than the thickness of human. To effectively heat ETAMs within the body, the heating distance must be extended to at least 100 mm. Additionally, the presence of bodily fluids (such as blood and mucus) in the conduits will enhance heat dissipation significantly more than in air.

2. Size of the heating area: The heating area of the ETAM is excessively large. From Figure 3D, it is evident that the size of the heating area approaches 1 cm. Moreover, as the distance from the coil increases, the high-frequency alternating magnetic field becomes more diffuse and its effective range expands, which could greatly affect selective heating. This not only limits the minimum lengths of the pre-designed P, F, and N region but also poses significant resistance to the precise segmentation of individual F or N region as described in the paper. Additionally, whether due to torsional, tensile, or bending forces, noticeable deformations occur at the fracture points. How might these deformations impact the alignment during subsequent merging processes?

3. Change in Overall Magnetization Direction: Once PCL is softened, it may become challenging to fix the internal permanent magnetic particles, resulting in these particles rotating under the influence of an external magnetic field rather than transmitting torque to the PCL matrix (Nat. Commun., 2020, 11, 6325). A detailed analysis of this phenomenon is necessary. On one hand, should the bending angles in Figure 2B take this into account? On the other hand, could this lead to unintended changes in the overall magnetic field direction?

4. The force applied for division: A detailed explanation of the division process depicted in Video S6 is needed, specifically regarding how forces are applied at both ends. The video should include not only an enlarged view of the fracture area but also overall experimental images. Furthermore, in Video S8, how the magnetic force for separation is applied should be demonstrated. What is the distance over which the force is applied, and does it meet the requirements for in vivo applications?

Comment #5 :

Regarding the capability of self-mergence, I have the following questions:

1. Alignment during self-mergence: How are the individual segments aligned during the self-mergence process? Specifically, in an in vivo setting, how is magnetic field guidance employed to achieve this alignment? The complex magnetic properties of ETAM present challenges for the independent control of already separated segments.
2. Predicting deformation: Given that the segments responsive to the magnetic field include both the F and N regions, what methods are employed to effectively predict the deformation of the continuum robot after self-mergence?

Minor issues:

Comment #6 :

Why is it necessary to preheat to 50°C for 30 minutes before cooling to room temperature in Figures 2G-H? What is the temperature and state of the continuum robot during the experiment? These aspects, which may lead to ambiguity, require a more detailed discussion for clarity.

Comment #7 :

During the manufacturing process, permanent magnets are placed beneath the mold to attract the ETAMs in the P and N regions to expel air bubbles. However, what is the approach for the P region, which does not contain magnetic particles?

Comment #8 :

How is the molten PCL uniformly mixed with the magnetic particles? When the viscosity is too high, it is challenging to achieve an even dispersion of the microparticles through stirring, while if the viscosity is too low, the microparticles tend to settle rapidly. Is it necessary to adjust the temperature of the PCL to achieve the optimal viscosity for effective mixing with the magnetic particles?

Reviewer #2

(Remarks to the Author)

The core concept introduced here is self-grafting. This is a really interesting idea shared with self-healing polymer materials and devices but shown here in a unique way for continuum magnetic robots.

Splitting of segments is done by thermally softening a region of the device and pulling or twisting it apart. Similarly, the device can be rejoined demonstrating high strength once re-joined. Overall I find the work interesting and unique. Although the feasibility for medical use is not clear to me, I think this work can be inspiring to further development of morphology-changing magnetic devices.

I don't understand how the twisting is accomplished in the demonstrations. It seems like it is done by hand in Fig 3E rather than magnetically? In particular, the twisting method seems difficult to do magnetically. The videos zoom in on the division site which hides the mechanism of pulling/twisting/bending in these demos. I see in the branch demo the division is done magnetically with much twisting.

The device is able to follow complex lumen networks in a desired pattern to varied targets as seen in Figure 2. This is impressive, although the magnetic field generation for these demonstrations is perhaps not realistic for the placement within the body as the field source is held directly under the workspace (5 mm gap) - unreasonably close to the working space. This allows for the generation of very high field gradients in a localized manner to pull the continuum along the desired trajectory. I don't see this as a fair demonstration, and would like to see how the device can target varied lumens with a field source at a distance outside the torso.

The localized heating is accomplished with an RF coil held immediately around the continuum with a distance of only a few cm at most. Could this heating be accomplished within the body distance? The manuscript should also comment on the safety of high power RF field generation within the body. Temperature safety is discussed in a reasonable way, but not field safety.

Very long thin tendrils of magnetic polymer are seen drawn out in the branching division video. These are not very visible in the figure 3G and not mentioned in the paper. Will these be acceptable clinically, or how could those be dealt with?

Graduated stiffness profiles in the flexible segments allows for reduced buckling when the continuum is pushed around corners. This seems like a good design choice.

Grasping using multiple divided and joined arms is very interesting. A dedicated grasping basket may be superior, but this is an interesting demonstration. Similar with the endoscope-mounted grasper.

I struggled to figure out what the P, N and F regions are. The explanation given needed a while for interpretation and I feel that the figures did not help understand this core concept.

P is the moderately-flexible part with no magnetic particles, and no heat-response.

N contains permanent magnet particles and are highly flexible. They experience a small amount of magnetic heating.

F is stiffness modulating by soft-magnetic (negligible magnetic coercivity), which respond to RF heating.

The language used is hard to follow at times. Ultimately I was able to find clarity, but it took sustained effort to read.

Version 1:

Reviewer comments:

Reviewer #1

(Remarks to the Author)

The authors have adequately addressed most of the previous comments. However, two major concerns remain unresolved. First, the term "self-graftable" appears to be misleading. The concept of self-graftability should encompass not only self-division and self-mergence but also inherent mobility characteristics—specifically, the ability to both separate autonomously and actively move to desired locations before merging with itself. This capability is fundamental to creating new structures and functions. Unfortunately, the manuscript fails to demonstrate such comprehensive grafting behavior. While the authors have included additional experiments showing the formation of a simple gripper, the gripper seems unreliable (moreover, the self-division process in this case requires clarification). In addition, Supplementary Material S21 only demonstrates the alignment of the facets to the facets, and not the alignment of the facets to other locations on the body. The authors should address whether it is feasible to achieve the structure and complete the task illustrated in Figure 6 through the claimed self-graftable properties.

Second, I am more concerned about whether the presence of body fluids or lubricants at the contact interface would impede self-mergence rather than self-division. For example, can the network structure shown in Supplementary Materials S24 be successfully achieved in a liquid environment?

Reviewer #2

(Remarks to the Author)

My prior concerns were primarily around the practicality of the proposed method. Overall I still have some significant concerns in that regard, but have been partially convinced. Some further specific concerns:

I still find the division method to be vague in the figure. It seems like a significant weakness of the concept that the splitting by magnetic field is quite limited and the demonstration (for example in Fig 3G) is requiring large bending back-and-forth. For the distance from the external permanent magnet to the lumen navigation experiments, you are now more clear that the distance is at least 10 cm for these demos. This is fine, it would be good to explain how much of the human body could be reached under such conditions.

Version 2:

Reviewer comments:

Reviewer #1

(Remarks to the Author)

The authors have adequately addressed all my previous concerns regarding the self-graftable properties and fluid environment performance. Their additional experiments convincingly demonstrate the technology's feasibility. I recommend this manuscript for publication in Nature Communications.

Reviewer #2

(Remarks to the Author)

I have no further comments.

Dear Editor and Reviewers,

We thank the reviewers for their valuable, constructive, and critical comments. We have addressed all of the comments in the revised version of the manuscript. A point-by-point response to the concerns and suggestions is provided here.

Reviewer #1:

The manuscript presents a novel design of FSCs that utilizes self-graftable capabilities—combining self-division and self-mergence—to navigate and function in confined and tortuous endoluminal scenarios. This is achieved through the development of ferromagnetic thermoplastic soft materials, which allow for reversible elastomer-fluid transitions, facilitating arbitrary division-mergence actions. The authors systematically explore the mechanisms underlying these functionalities and demonstrate the practical applications of this technology in performing complex medical tasks like targeted delivery, foreign body removal, and thermal ablation in simulated environments. Regarding the experimental demonstration, robot design, self-division, and self-mergence, the reviewer should pose the following concerns.

[Comment 1-1]

It is important to note that the dimensions of continuum robots are determined by the size of the endoluminal scenarios in which they operate. Robots used in neurointerventional or cardiac interventional surgeries require very small sizes, which results in the medical devices they carry being less sophisticated. In contrast, endoscopies intended for the respiratory, digestive, and urinary systems are not subject to such stringent size requirements. There are many endoscope-compatible medical devices available for these systems, capable of performing tasks such as foreign body retrieval, lithotripsy, electrosurgery, and radiofrequency ablation, all of which have been clinically tested. It is necessary to limit the scope of application of the proposed continuum robots.

Utilizing smaller robots can reduce invasiveness and enhance patient-friendliness, but this naturally presents challenges in functional integration. If it is possible to reduce size while maintaining reliability and functionality, that would represent a significant advancement. The grafting functions proposed by in this work are an effective potential method for functionalizing small-sized robots. However, three experimental demonstrations presented in the manuscript predominantly showcase the application of controllable adhesion, and do not effectively demonstrate the claimed division and merger functions. Beyond connections, division and merger also require controlled movement of both ends. For instance, navigating a robot through a confined lumen to an open space, where it can self-divide into multiple untethered segments that merger into a functional structure larger than the orifice to perform tasks, is beyond the capability of existing robots and can strongly support the arguments made.

[Response 1-1]

We thank the Reviewer's careful comments and suggestions on the application scope, which helped us to improve the demonstration.

We agree with the Reviewer that limiting the application scope of the proposed ETACs is necessary. We aim to demonstrate three fundamental properties of ETACs: 1) small size for navigating confined

spaces (Figures 2G-H), 2) sufficient flexibility for operational tasks (Figures 2G, 5, and 6), and 3) self-grafting ability to create new structures/functions (Figures 5-7). Therefore, we selected different application scenarios to highlight these three properties accordingly (please refer to the detailed explanation in Table R1-1). Meanwhile, we agree with the Reviewer that we should present a scenario showing a comprehensive demonstration of all ETAC's properties. For example, a robot navigates through a confined lumen to an open space, allowing it to self-divide into multiple sub-continua that merge into a functional structure larger than the orifice to perform a task.

Table R1-1. Detailed explanation of the presented demonstrations.

Demonstration	Aim	Reason	Scalable?
Continuum carriers	 Showcase self-grafting Showcase flexibility 	We show ETAC's self-grafting ability and adjustable flexibility by setting up a 3-lumens phantom.	√ (Can be scaled down to mm-level)
Airway foreign body removal	 Showcase self-grafting Showcase flexibility 	We show ETAC's self-grafting ability by using a grasper to remove foreign bodies. The grasper can be assembled in vivo or before conducting an experiment.	√ (Can be scaled down to submillimeter-level, such as bronchus)
RF ablation	 Showcase self-grafting Showcase tiny size (in vivo animal test) 	With self-grafting, the remote control of multiple segments can be arranged in order. ETACs can complete ablation tasks in more confined spaces due to their tiny sizes.	√ (Already in millimeter-scale but can scale up to centimeter level)
In vivo meshing	 Showcase self-grafting Tiny size Flexibility 	We newly added a demonstration Showcasing all of ETAC's fundamental properties, as suggested by the Reviewer.	√ (Can be scaled down to submillimeter-level)

As shown in Figure R1-1, we have newly designed a demonstration to prove that ETACs can complete the task as the Reviewer suggested. This demonstration aims to showcase that ETACs can navigate through a confined and curved lumen to an open space, where the ETAC can operate in vivo meshing for varied purposes, such as in situ tissue protection and in situ stent repairment [R1-R2]. The ETAC navigated through a confined, curved lumen controlled by magnetic fields (Figure R1-1 i). It is accurately targeted at a point on the inner wall of the open space, where the ETAC's tip can attach. An RF field was applied at the attachment region to soften the ETAC tip for adhesion (Figure R1-1 ii). After cooling the attachment region for a few seconds, another RF field was applied to the end of the exposed ETAC for self-division (Figure R1-1 iii). This process generated an untethered segment with one end bonded to the environment. By guiding the other end to the opposite target and conducting another RF merge, the untethered segment can be fixed on the environment (Figure R1-1 iv). By repeating the

abovementioned steps multiple times, the in vivo meshing can be realized (Figures R1-1 v-vi). This demonstration also suggests the possibility of in vivo printing, which follows a similar concept to the in vivo meshing.

Regarding the other three demonstrations (continuum carriers, airway foreign body removal, and aided submucosal dissection), we note that the scenario dimensions are scalable. The dimensions of the experimental setup can be scaled down to the millimeter level and ETAC's diameter can be easily reduced to less than the millimeter scale. Given the abovementioned properties, ETAC can also be utilized to conduct surgeries in animals with smaller body sizes, such as the mice RF ablation demonstration shown in Figure 8.

According to the Reviewer's suggestions, we present potential biomedical applications in Figure R1-2. Given the self-grafting function, ETACs hold the potential to conduct procedures requiring passing through narrow bottlenecks, such as endoscopic retrograde cholangiopancreatography and transurethral procedures, which the common functional continuum cannot easily insert. ETACs also have the potential to operate minimally invasive surgeries for small-size animals, such as nasal passage intubation, RF ablation in the gastrointestinal tract (demonstrated in Figure 8), and respiratory system navigation. The abovementioned lumens are of narrow sizes ranging from 1 mm to 5 mm, which are challenging for existing continuum robots to navigate and operate complex tasks.

We have supplemented the new demonstration to the main text and the changes are highlighted in yellow on Page 4 Column-1 Ln 20-24, Page 17 Column-2 Ln 11-22, and Supplementary Materials S24.

Figure R1-1. In vivo meshing. (i) ETAC navigated through a confined and curved lumen (with an inner diameter of 1 mm) to an open space. (ii) ETAC was magnetically guided to aim at the target point and approach by insertion. (iii) Self-division and self-mergence were used to bond the tip to the environment and split it into an untethered segment. (iv) Untethered merging with the external environment. (v) In vivo meshing can be realized by repeating multiple times. (vi) Zoom-in view of the formed mesh.

Figure R1-2. Biomedical application scopes of ETACs requiring passing through narrow bottlenecks. Human procedures include endoscopic retrograde cholangiopancreatography and transurethral procedures. Small-size animal procedures include nasal passage intubation, RF thermal ablation in the gastrointestinal tract, and respiratory system navigation.

[Comment 1-2]

For the first demonstration (Movie S10, Continuum Carriers: Division and Mergence): The author demonstrates how by breaking and merging, the robot is endowed with three tips, allowing it to remove foreign bodies from multiple channels simultaneously. The endoluminal environment is sufficiently large to accommodate the functional structure formed by the robot. In such large channels, would it be feasible to manually assemble this structure externally before commencing the task, or could directly inserting three separate continuum robots at the junction to perform tasks independently achieve quicker results?

[Response 1-2]

We thank the Reviewer’s comments on the “Continuum Carriers” demonstration. This demonstration aims to show ETAC’s potential to conduct self-grafting in multiple channels and grasping capability by adapting various geometries with enhanced flexibility. We agree with the Reviewer that the channels are relatively large and the lumen size can be scaled down. Combining the Reviewer’s concern on

“manually assemble this structure externally before commencing the task, or could directly inserting three separate continuum robots at the junction or perform tasks independently achieve quicker results”,

we believe ETACs can advance over other devices. Regarding the current channel size (Diameter =1.5 cm), inserting three separate continuum robots simultaneously is feasible. However, as the Reviewer mentioned, when there is a bottleneck terrain during continuum robots’ navigation to the target region, simultaneously inserting multiple continuum robots will be challenging to realize (please refer to the bottleneck scenario applications in Respons 1-1 and Figure R1-2). Moreover, the number of simultaneous inserted continuums is limited by the size of the main channel. For example, when the sub-channel numbers are increased to a large number (e.g., 10 sub-channels), the main channel may have insufficient size (requiring ~1-2 cm diameter) to hold too many continuum robots to insert simultaneously.

[Comment 1-3]

2. For the second demonstration (Movie S11, Graftable End Effector: Airway Foreign Body Removal): The robot is formed into a grasper, and effective capture is achieved using the adhesive properties of PCL. Was this grasper assembled in situ or manually assembled externally? What are the advantages over existing bronchoscopic forceps?

[Response 1-3]

The grasper shown in the Supplementary Movie S11 was assembled externally by manual approach. We thank the Reviewer's reminder on this demonstration that the grasper can be assembled in situ. Therefore, we have revised the Supplementary Movie S11 by adding an in situ assembling process, as shown in Figure R1-3.

Due to the flexibility and self-grafting ability, an ETAC can be in situ assembled through five steps. An ETAC was first inserted into a trachea and magnetically guided to the target region (Figure R1-3 B-i). By applying a magnetic field in the opposite direction of ETAC's Preprogrammed magnetization profile in the axial direction, the ETAC was folded to form a circle shape at the tip region (Figure R1-3 B-ii). After aligning, an RF field was applied at the tip region to conduct self-mergence (Figures R1-3 B-iii&iv). Then, the tip of the circle region was heated to conduct self-division, forming a two-arm grasper (Figures R1-3 B-v). The assembled grasper can perform the same functions as demonstrated previously (Figure R1-3 C).

Figure R1-3. In situ assembly. (A) There are five steps to assemble an ETAC into a grasper, including navigation to the target, magnetic folding, aligning, self-mergence, and self-division. (B) Real-time figures showing the assembling process in an ex vivo pig trachea. (C) The in situ assembled grasper can conduct the same functions as shown in the previous demonstration, such as navigation, grasping, and releasing.

Regarding the advantages over other existing bronchoscopic forceps, as stated in the main text, there is a risk of unsuccessful object grasping with existing devices for foreign body removal due to their grasping principle [R3-R5]. Two primary methods are commonly employed to accomplish tasks. One method uses retrieval baskets, which constrain objects by applying mechanical forces. The retrieval

baskets are utilized to remove large objects with irregular geometries. However, during this process, there is a possibility of the object escaping due to the counterforce exerted on the basket, which may lead to further issues such as relocation or even severe channel blocking [R3]. Moreover, inserting retrieval baskets into confined body lumens can result in side effects such as mucosal injury. The other approach is using endoscopic graspers, which have the limitation of adapting to oversized or fragile objects such as nuts [R4-R5]. This work provides an alternative approach to complete airway foreign body removal with minimal object disturbances. The proposed ETAM-grasper has the potential to grasp irregular large geometries while preventing damage to fragile objects.

We have added the corresponding descriptions in the main text and the changes are highlighted in yellow on Page 14 Column-2 Ln 25-26 and Supplementary Materials Section 13.

[Comment 1-4]

3. For the third demonstration (Movie S12, Assisted Submucosal Dissection"): How are the two continuum robots independently operated? Considering the intestinal wall as a working environment in a natural orifice transluminal endoscopic surgery, arranging these two instruments in an appropriate posture at the surgical site is a non-trivial issue. Particularly since both tools are fine, have an elastic modulus only in the MPa range, and are magnetic, significant mutual interference cannot be ignored. What is the movement precision of the ETAM aiding the ESU?

[Response 1-4]

We interpret the Reviewer's comments to two main questions: 1) How are the two continuum robots independently operated in Supplementary Movie S12 and 2) What is the movement precision of the ETAM aiding the ESU?

Regarding the first question, we specially designed the ESD process into 3 steps (thermal ablation, pulling assistance, and dissection) to separately operate the two continuum robots without mutual interference. In the first step, the ETAC can navigate to the target region assisted by an endoscope and conduct thermal ablation to the target region (Figure R1-4). After thermal ablation, the ablated tissue can be firmly bonded to the ETAC's tip. This step was proven in both Supplementary Movie S9 and S12. The ESU was placed far from the magnetic field during this step. Then the ESU can be guided to the target region while the ETAC for ablation can lift the tissue by manually pulling the remote end. Due to the stronger pulling force compared to the magnetic force, when the ESU is conducting dissection, the ablation ETAC will not be affected by the control magnetic field.

Figure R1-4. An ETAC equipped with an endoscope.

Regarding the movement precision of the ETAM aiding the ESU, we have conducted a series of movement precision tests. In the continuum control area, there are two main test methods to determine the moving precision of a continuum robot: movement resolution and position repeatability. As shown in Figure R1-5 A, we installed an ETAM-ESU in a Helmholtz coil where a grid mesh was placed beneath

the ESU for motion measurement. The magnetization profile of the ETAM-ESU was preprogrammed along the axial direction and the magnetic field direction was perpendicular (defined as y-direction) to the magnetization profile for bending actuation. The movement resolution results are shown in Figures R1-5 B-C. The ETAM-ESU performs satisfactory linear controllability with the increment of actuation voltage (namely magnetic moment). As shown in Figure R1-5 D, the repeatability showed $\sim 0.3^\circ$ averaged orientation error and ~ 0.6 mm averaged positioning error. It was observed that the position repeatability is higher in the low magnetic moment range than in the high moment range. This is because the positioning error accumulates with the increment of the magnetic moment.

We have added the corresponding description to the main text; the changes are highlighted in yellow on Page 20 from Column-1 Ln 34 to Column-2 Ln 5 and Supplementary Materials S14.

Figure R1-5. Movement precision test of ETAM-ESUs. (A) Experimental setup of movement precision. (B) Real-time image of the movement resolution test results. (C) Movement resolution results showing a linear control relation between actuation required voltages and Y-direction displacement. (D) Position repeatability results showing a relation between the actuation magnetic field strength and bending performances.

[Comment 2-1]

In reviewing the design aspects of the robot as presented in the paper, the following concerns are raised:

1. Temperature monitoring of PCL: Since PCL is not encapsulated, maintaining its temperature below the melting point is crucial to ensure the stability of its form. How does the study monitor the temperature of PCL to guarantee the structural stability and safety of the surgical procedure, particularly given the challenges associated with measuring the temperature of robots in body using infrared thermography?

[Response 2-1]

We thank the Reviewer's comments on the robot's design, from which we significantly benefit to improve our work quality. Although PCL is not encapsulated, it can maintain its temperature below the melting point, and the melting temperature can be modified by selecting PCL raw materials and adjusting raw material mixing ratios. Given our test experiences, ETAM's overall melting point can be adjusted from 40 °C to 100 °C. Moreover, we agree with the Reviewer that it is necessary to monitor the device's temperature to guarantee structural stability and safety during the surgical procedure. Here, we propose an empirical temperature monitoring method for future detailed studies.

The device temperature changes with multiple parameters, such as heating time, the RF coil distance to the device, and the environmental conditions (mucous in lumens). Considering all these factors, we have designed an experiment for temperature monitoring. As shown in Figure R1-6 A, we installed a thickness-adjustable holder on an RF coil and an ETAM was placed at the middle of the holder. An infrared camera was utilized to record the temperature changes at the top of the holder (Figure R1-6 B). The temperature was collected and plotted as shown in Figure R1-6 C. When the heating distance is close to the ETAM (less than 1 cm, which is impractical for practical application), the temperature significantly increases in less than 1 minute. In such cases, although the relation between heating time and temperature is linear, the over-sensitive temperature changes will be difficult to control for operational safety. With an increment in heating distance, the heating speed tends to slow down and become more controllable (such as at 12-16 cm heating distance). Nonlinear curve fitting can be adopted to establish an empirical formula to predict the temperature change. Moreover, lubrication from mucous was observed to slow down the heating speed due to heat dissipation, which can also be considered as an influence parameter to be included in the empirical formula.

We have added the corresponding descriptions to the Supplementary Materials S15.

Figure R1-6. Empirical temperature monitoring method. (A) Experimental setup for temperature monitoring test. (B) Obtained infrared image for data collection. (C) Experimental results showing relations between heating time, device temperature, heating distance, and lubrication status.

[Comment 2-2]

2. Lubrication needs: The demonstrated robot lacks a lubricating coating or lubricant on its surface. However, lubrication is essential during the deployment of continuum robots, not only to facilitate intubation but also to minimize tissue damage (Adv. Material., 2019, 31, 1807101). In clinical settings, lubrication is also an indispensable operation. Could lubrication enhance the performance of the proposed continuum robot, for example, by facilitating the smooth navigation of the robot with a uniform elastic modulus as shown in supplementary Movie S2? Moreover, would the use of lubricating coatings or lubricants impact the self-graftable capabilities of PCL within the ETAC? And considering the in vivo experiments conducted on mice, it is important to assess how the naturally secreted fluids within bodily

conduits, which provide lubrication, might affect the self-graftable capabilities of the robot.

[Response 2-2]

We agree with the Reviewer that lubrication is necessary for most continuum robot-assisted surgeries to minimize tissue damage. We interpret the Reviewer's comments into three questions: 1) Could lubrication enhance the performance of uniform modulus ETACs? 2) Will the self-grafting function be affected by lubrication? 3) How do the natural fluids within bodily conduits affect self-grafting function?

Figure R1-7. Comparison test between uniform modulus ETAC, uniform modulus ETAC with lubrication, graded-stiffness ETAC, and graded-stiffness ETAC with lubrication.

Regarding the first question, we have newly conducted a comparison test between uniform modulus ETAC, uniform modulus ETAC with lubrication, graded-stiffness ETAC, and graded-stiffness ETAC with lubrication, as shown in Figure R1-7. To maintain ETACs' simple structural design and self-grafting normal function, we applied medical device lubricants as instructed by medical doctors (No. LUB0005, Health&Beyond Hygienic Product Inc., China) instead of using lubrication coating layers. The results show that the lubrication did not improve a uniform modulus ETAC's navigation performance. The uniform modulus ETAC failed to complete a $\sim 270^\circ$ steering due to the unoptimized force transmission, which is not closely related to the lubrication condition. When the graded stiffness structure is applied to ETAC's tip, no obvious differences were observed between ETACs with/without lubrication. Compared with the silicon rubber-based or TPU-based continuum robots, PCL materials have relatively lower surface friction, which may reduce the possibility of tissue damage.

Regarding the second and third questions, we have newly conducted a qualitative experiment and a quantitative test to investigate the heat dissipation effect of lubrication. As shown in Figure R1-8 A, we installed an ETAC inside an RF coil with a weight attached at the end of the ETAC. The ETAC completed self-division in ~ 500 ms after turning on the RF machine (Figure R1-8 B). We further installed an ETAC inside a tube filled with water for comparison (Figure R1-8 C). No obvious differences were observed with respect to the division shapes and time (Figure R1-8 D). This is because the heating power is relatively high and the heating distance is short enough to minimize the performance differences. The results indicate that improving the heating efficiency will minimize the heat dissipation effect from lubrication. Results shown in Figure R1-6 provide a quantitative comparison between ETACs with and without lubrication. For practical uses, heat dissipation can be systematically tested to develop a temperature monitoring empirical formula.

We have added corresponding descriptions in the main text and Supplementary Materials. The

changes are highlighted in yellow on Page 7 Column 2 Ln 18-29 and Supplementary Materials S16.

Figure R1-8. Qualitative comparison tests on the heat dissipation effect of lubrication. (A) Experimental setup of the non-lubrication test. (B) Self-division of the non-lubrication test. (C) Experimental setup of the lubricated test. (D) Self-division of the lubricated test.

[Comment 2-3]

3. *Material state of ETAM: PCL's elastic modulus decreases with increasing temperature (Soft Robot., 2022, 9, 189-200). The mechanical properties of ETAM should be given specific values to differentiate between the three states (the elastomer, softening, and fluid state). Considering the heat dissipation of liquids in clinical applications, could this lead to changes in the elastic modulus of continuum robots, affecting the numerical control of tip.*

[Response 2-3]

We agree with the Reviewer that providing specific values to differentiate between the elastomer, softening, and fluid states is necessary. Given ETAM's wide material modifying space, hundreds of ETAMs could be generated with different specific values for softening and fluidification. Instead of quantifying every type of ETAMs, we would like to provide a quantification principle to guide engineers in customizing ETAMs.

The Young's modulus and the viscosity value are respectively defined as the softening point and fluidification point. Based on test experiences, when the ETAM stiffness is reduced below 1 MPa (namely heated to the kPa level), it can be divided with a force lower than 2 N. Regarding the fluidification point, 440 Pa·s is the highest viscosity we collected where liquid ETAM can conduct flowing motions under magnetic actuation, which is defined as the fluidification point. It should be noted that different ETAMs require varied temperatures to achieve the above softening and viscosity values. By recording the softening and fluidification temperatures of different ETAMs, a material property database can be established. Here, we provide preliminary results in Figure R1-9 and Table R1-2.

We would like to note that the specific values are only for illustrating the quantification principle, which varies with different practical needs. For example, if the surgical condition can support higher division forces, the softening point will be improved accordingly. Therefore, utilizing the collected testing data to develop an empirical formula, as we stated in Response 2-1 will be more valuable, which will also involve the heat dissipation factor from lubrication or bodily fluids.

We have added the corresponding descriptions to Supplementary Materials S17.

Figure R1-9. Relation between ETACs' stiffness, MNP proportions, ETAC types, and temperature.

Table R1-2. Detailed viscosity values with respect to different cut rates, temperatures, and material types.

Material No.	Temp. (°C)	RPM	Cut rate	Viscosity (mPa*s)	Cut pressure (mPa)
F-ETAC-100%	140	5	1.2	361410.1	451762.7
F-ETAC-100%	140	6	1.5	342579.6	513869.4
F-ETAC-100%	140	7	1.7	327832.8	573707.4
F-ETAC-100%	140	8	2	305712.6	611425.2
F-ETAC-100%	140	9	2.2	312581.8	703309.2
F-ETAC-100%	145	5	1.2	331916.5	414895.6
F-ETAC-100%	145	6	1.5	306468.8	459703.3
F-ETAC-100%	145	7	1.7	287157.5	502525.7
F-ETAC-100%	145	8	2	275368.2	550736.4
F-ETAC-100%	145	9	2.2	280945.5	632127.5
F-ETAC-100%	150	5	1.2	290852.3	363565.4
F-ETAC-100%	150	6	1.5	265820.6	398730.9
F-ETAC-100%	150	7	1.7	231411.4	404969.9
F-ETAC-100%	150	8	2	234105.5	468211

F-ETAC-100%	150	9	2.2	224983.3	506212.4
F-ETAC-100%	155	5	1.2	249561.3	311951.6
F-ETAC-100%	155	6	1.5	229520.8	344281.2
F-ETAC-100%	155	7	1.7	218771.2	382849.7
F-ETAC-100%	155	8	2	213261.4	426522.9
F-ETAC-100%	155	9	2.2	204060.5	459136.1
F-ETAC-100%	160	5	1.2	228462	285577.5
F-ETAC-100%	160	6	1.5	199838.1	299757.1
F-ETAC-100%	160	7	1.7	204672.6	358177.2
F-ETAC-100%	160	8	2	196245.9	392491.8
F-ETAC-100%	160	9	2.2	203556.3	458001.7
N-ETAC-100%	140	5	1.2	115025	143781.3
N-ETAC-100%	140	6	1.5	111735.4	167603.1
N-ETAC-100%	140	7	1.7	111168.2	194544.4
N-ETAC-100%	140	8	2	110884.6	221769.2
N-ETAC-100%	140	9	2.2	110538	248710.5
N-ETAC-100%	145	5	1.2	94833.3	118541.6
N-ETAC-100%	145	6	1.5	95097.9	142646.9
N-ETAC-100%	145	7	1.7	93666.5	163916.4
N-ETAC-100%	145	8	2	93443.6	186887.3
N-ETAC-100%	145	9	2.2	93144.3	209574.7
N-ETAC-100%	150	5	1.2	85077.7	106347.1

N-ETAC-100%	150	6	1.5	81674.6	122511.9
N-ETAC-100%	150	7	1.7	80054	140094.6
N-ETAC-100%	150	8	2	79405.8	158811.7
N-ETAC-100%	150	9	2.2	79027.7	177812.4
N-ETAC-100%	155	5	1.2	69423.4	86779.2
N-ETAC-100%	155	6	1.5	68251.2	102376.8
N-ETAC-100%	155	7	1.7	67738	118541.6
N-ETAC-100%	155	8	2	67353.1	134706.3
N-ETAC-100%	155	9	2.2	67305.9	151438.3
N-ETAC-100%	160	5	1.2	60121.5	75151.9
N-ETAC-100%	160	6	1.5	58798.1	88197.2
N-ETAC-100%	160	7	1.7	58663.1	102660.4
N-ETAC-100%	160	8	2	58420	116840
N-ETAC-100%	160	9	2.2	58230.9	131019.6

[Comment 3]

The clarification regarding the type of magnetic field device employed in Movie S3 is necessary—was it a coil or a permanent magnet? Particularly in the navigation to target 19, there are two locations prone to buckling. Wrapping a helix around the continuum robot can prevent buckling by transferring thrust from the rear to the tip (Sci. Robot., 2024, 9, eadh0298), how was buckling avoided in this study?

[Response 3]

We agree with the Reviewer that it is necessary to clarify the type of employed magnetic field device in Supplementary Movie S3. The previous gradient field is generated by a permanent magnet. This comment reminds us that the presentation of both permanent magnet and Helmholtz coil actuation may be needed. Therefore, as shown in Figure R1-10, we have supplemented a complex channel navigation demo realized by a Helmholtz coil. We also optimized the permanent control distance for practical uses by improving the permanent magnet strength and ETAC's magnet particle content, resulting in an actuation distance of more than 10 cm from the ETAC tip (Figure R1-10 A).

We installed the phantom in a Helmholtz coil (Figure R1-10 B) and successfully realized the varied target navigation, which was previously realized by permanent magnet control (Figures R1-10 C-D). It

proves that ETACs can be accurately controlled to conduct navigations by both external permanent magnetic field (gradient) and uniform electromagnetic magnetic field.

The Reviewer also raised concerns about the buckling phenomenon when ETAC navigated to target 19 in Supplementary Movie S3. In our previous demonstration, our main idea was to showcase ETAC's satisfactory navigation ability with the graded-stiffness tip design. However, the buckling phenomenon indeed existed. Optimizing the graded-stiffness arrangement can avoid buckling during navigations. By softening the flexible region and hardening the rigid region, we successfully avoided buckling without adding more structures (such as wrapping a helix as the Reviewer suggested), thereby maintaining ETAC's simple structure. As shown in Figure R1-11 A, multiple buckling cases were observed during a homogenous (without graded-stiffness tip design) ETAC's navigation to target 19, because the force from the remote end cannot be effectively transmitted to the tip. By using our optimized ETAC with the graded-stiffness tip, the buckling was successfully eliminated (Figure R1-11 B). Considering the Review's concern about the lubrication issue in Comment 2-2, we also tested the graded-stiffness ETAC with lubrication, which also successfully reached target 19 without buckling (Figure R1-11 C).

Figure R1-10. Comparisons between the permanent magnet control and electromagnetic control. **A** Experimental setup of the external permanent magnetic (EPM) actuated navigation. **B** Experimental setup of the electromagnetic actuated navigation. **C** Previous results of EPM actuated navigation. **D** Results of electromagnetic navigation.

Figure R1-11. Performance comparisons between homogenous ETAC, graded-stiffness ETAC, and graded-stiffness ETAC with lubrication. **A** Multiple buckling cases were observed during homogenous ETAC’s navigation to target 19. **B** No buckling was observed during the optimized graded-stiffness ETAC’s navigation to target 19. **C** No obvious differences were observed by comparing the performances between graded-stiffness with and without lubrication.

We have added a corresponding description in the main text to help the reader understand. The changes are highlighted in yellow on Page 7 Column 2 Ln 13-17 and Supplementary Materials S16.

[Comment 4-1]

About the self-division capability of the proposed system, the following concerns are raised:

1. Heating efficiency of ETAM: The heating efficiency of ETAM decreases sharply as the distance from the coil increases. As shown in Fig. 3C, a distance of 25 mm significantly reduces heating efficiency, which is much less than the thickness of human. To effectively heat ETAMs within the body, the heating distance must be extended to at least 100 mm. Additionally, the presence of bodily fluids (such as blood and mucus) in the conduits will enhance heat dissipation significantly more than in air.

[Response 4-1]

We understand the heating distance may still not be practical for some specific scenarios, such as in the digestive and respiratory systems. We regard this issue as an industrial problem that will be effectively addressed by increasing the size and power of the RF machine [R6].

Here, we provide our rationale regarding the concerns on heating efficiency and resulting field safety in detail. Current laboratory RF equipment is limited by its size and power, which are much lower than those of existing industrial and medical devices [R6-R8]. For example, the RF frequency of existing medical devices ranges from 128 MHz (3T) to 342 MHz (8T), which is more than 400 times higher than our laboratory RF machine [R6]. While providing satisfactory RF efficiency, such industrial RF machines can support relatively long-distance heating. The maximum coil diameter we found is 56 cm, which is acceptable for practical application [R7-R8]. Regarding the field safety consideration, there exists animal safety evaluation research for extreme RF exposure. Researchers found that 10–12 T RF exposure for a continuous 28 days was relatively safe for mice. It was also reported that 3.5–23.0 T RF exposure for 2 h and 7.0–33.0 T for 1 h did not have severe long-term detrimental effects on mice [R9-R10]. Therefore, we regard heating efficiency and field safety as well-proven issues that can support our proposed self-grafting technique.

The changes are highlighted in yellow on Page 21 Column-1 Ln 20-42.

[Comment 4-2]

2. *Size of the heating area: The heating area of the ETAM is excessively large. From Figure 3D, it is evident that the size of the heating area approaches 1 cm. Moreover, as the distance from the coil increases, the high-frequency alternating magnetic field becomes more diffuse and its effective range expands, which could greatly affect selective heating. This not only limits the minimum lengths of the pre-designed P, F, and N region but also poses significant resistance to the precise segmentation of individual F or N region as described in the paper. Additionally, whether due to torsional, tensile, or bending forces, noticeable deformations occur at the fracture points. How might these deformations impact the alignment during subsequent merging processes?*

[Response 4-2]

We understand the Reviewer’s comment on the size of the heating area. We propose two methods to reduce the distance effect on the heating area: 1) customize RF coil geometry and 2) utilize a heat distribution center to conduct self-grafting.

The first approach is customizing RF coil geometries to fit different conditions. The width and depth of generated RF fields strongly depend on the geometries of RF coils. As shown in Figures R1-12 A&B, for example, the solenoid coil generates long depth and small width RF fields, which fit for long-distance heating scenarios. However, disk coils generate short depth and large width fields, which fit for wider area heating when there are multiple regions to be heated. We understand that only adjusting coil geometries may still face division precision issues. Therefore, adopting our proposed second method will further reduce the heating area effect and improve the division precision.

The second method is utilizing the heat distribution center to conduct self-grafting. Given the RF field’s gradient property, every RF field has a central region providing the strongest heating effect. We regard this region as the heating center (Figure R1-12 C). It is true that the RF field becomes more diffuse as the distance from the coil increases. However, the heating center still exists and its position can be adjusted by moving coils. After a few seconds of cooling, the heating center remains in the softened state while other heated regions are at a lower temperature. At this time, the heating center is the place to conduct self-grafting, including division and mergence. This approach has also been proven to be feasible in Response 5-1. Regarding the alignment concern, we would like to respond together with Comment 5-1 as these concerns are similar. Please kindly find the detailed explanation in Response 5-1.

We have added the corresponding descriptions in the main text and Supplementary Materials. The changes are highlighted in yellow on Page 10 Column-2 Ln 14-17 and Supplementary Materials S18.

Figure R1-12. Two methods to reduce heating area: customizing RF coil geometry and utilizing heat distribution center. (A) The RF field generated by a solenoid coil. (B) The RF field generated by a disk coil. (C) Infrared images showing the heating center's existence.

[Comment 4-3]

3. Change in Overall Magnetization Direction: Once PCL is softened, it may become challenging to fix the internal permanent magnetic particles, resulting in these particles rotating under the influence of an external magnetic field rather than transmitting torque to the PCL matrix (*Nat. Commun.*, 2020, 11, 6325). A detailed analysis of this phenomenon is necessary. On one hand, should the bending angles in Figure 2B take this into account? On the other hand, could this lead to unintended changes in the overall magnetic field direction?

[Response 4-3]

It is true that magnetic particles have the chance to rotate under an external magnetic field rather than transmitting torques when the PCL matrix is significantly softened. However, such a phenomenon was not obviously observed in our case due to the following reasons:

- 1) The magnetization re-arrangement phenomenon strongly depends on the softening condition of the PCL matrix. It is known that there is a temperature range greater than 40 degrees Celsius for the softening state, which is a transitional state between the elastomer state and the fluid state. The temperature we chose to conduct actuation is located at the lower region of the softening state temperature range. Under such conditions, the PCL matrix has a relatively satisfactory ability to transmit torque.
- 2) In Figure 2B, the tiny errors between heating and after curing the ETACs are mainly due to the stiffness differences, which are also well-proven by numerical simulations. The cure time we selected is 10 hrs for 86% - ~100% stiffness recovery rate. Given our experiences, within 24 hrs curing, there are no obvious bending angle differences between the original ETAC and cured ETAC.
- 3) We also conducted magnetization profile measurements to further support our claim. As shown in Figure R1-13 A, an ETAC was magnetized in the vertical direction and five measurement points were selected to record the magnetic strength changing before and after bending actuation under the softening state. Figure R1-13 B shows that there are no obvious changes in magnetic strength observed.

We have added the corresponding descriptions to the main text and Supplementary Materials. The changes are highlighted in yellow on Page 10 Column-1 Ln 10-17 and Supplementary Materials S19.

Figure R1-13. Magnetic strength comparisons. (A) Experimental principle. (B) Quantitative comparison

results.

[Comment 4-4]

4. The force applied for division: A detailed explanation of the division process depicted in Video S6 is needed, specifically regarding how forces are applied at both ends. The video should include not only an enlarged view of the fracture area but also overall experimental images. Furthermore, in Video S8, how the magnetic force for separation is applied should be demonstrated. What is the distance over which the force is applied, and does it meet the requirements for in vivo applications?

[Response 4-4]

We agree with the Reviewer that the demonstration of Supplementary Movie S6 is unclear regarding how forces are applied at both ends. Therefore, we have revised the Supplementary Movie S6 and provided further descriptions.

As described in the main text, self-divisions can be completed manually (Figure 3E) or magnetically (Figure 3G). Figure R1-14 A presents manual and magnetic division comparisons for the Reviewer's information. The manual approach is simple and direct, while the magnetic approach shows apparent advantages in performing remote division in confined lumens (Figure R1-14 B). The applied forces are marked in the figure. We note that the magnetic twisting is realizable when the holding force is sufficient by a stronger magnetic field strength.

Regarding the force application distance at the magnetic tip end, the control distance strongly depends on the magnetic field strength generated by the actuation device. According to our optimized test results (we optimized the permanent control distance for practical uses by improving the permanent magnet strength and improving ETAC's magnet particle content, resulting in an actuation distance more than 10 cm from the ETAC tip), the control distance for self-division ranges from 5 cm to 12 cm, which is suitable for special needs in biomedical application scenarios.

We added the corresponding descriptions to Supplementary Materials S20 to clarify our statement.

Figure R1-14. Comparisons between the manual and magnetic division approach. (A) (i-ii) pulling division, (iii-iv) bending division, and (v) twisting division. (B) Application scenarios for manual and magnetic divisions.

[Comment 5-1]

Regarding the capability of self-mergence, I have the following questions:

1. *Alignment during self-mergence: How are the individual segments aligned during the self-mergence process? Specifically, in an in vivo setting, how is magnetic field guidance employed to achieve this alignment? The complex magnetic properties of ETAM present challenges for the independent control of already separated segments.*

[Response 5-1]

The alignment issue reveals ETAC's another advantage of conducting self-alignment due to the continuum's nature. We have conducted a series of division-mergence tests to showcase the self-alignment performances. Due to the continuous magnetization distribution, the divided segments can be regarded as multiple magnets that maintain their original magnetization profiles. Regardless of the division methods adopted (Figure R1-15) or the geometries at the division region, by bringing the two division ends close to a short distance, the divided two segments can attract and align with each other. We also observed that the two segments cannot easily align with each other due to environmental friction. This issue can be addressed by adjusting the main segment's position by manually pushing/pulling its remote end to shorten the distance between the two segments. During this process, no external magnetic field is required.

We have added the corresponding descriptions to the main text and Supplementary Materials. The changes are highlighted in yellow on Page 11 Column-2 Ln 1-9 and Supplementary Materials S21.

Figure R1-15. Self-alignment of separated segments. (A) Self-alignment of manually divided segments. (B) Self-alignment of magnetically divided segments after RF heating. (C) Self-alignment of manually divided segments after RF heating. (D) Self-alignment of magnetically divided segments after an RF heating-cooling loop.

[Comment 5-1]

2. *Predicting deformation: Given that the segments responsive to the magnetic field include both the F and N regions, what methods are employed to effectively predict the deformation of the continuum robot after self-mergence?*

[Response 5-1]

The deformation prediction after ETACs' self-mergence is convenient and direct, which is an advantage

of ETAC's simple structure design. As described in the main text, F-regions and N-regions are respectively employed to conduct division/mergence tasks and deformation tasks, due to the RF-responsive capability of F-regions and the dipolar magnetic performances of N-regions. When F and N-regions are simultaneously under an external magnetic field, F regions tend to be attracted under the gradient magnetic field direction, while N-regions perform dipolar performances with respect to the field direction. Such deformations are directly predicted by numerical simulations.

To respond to the Reviewer's comment, we newly calculated 3 cases for comparisons and illustrations. As shown in Figure R1-16, we preset 3 designs of self-merged structures with different magnetic profiles. Our developed FE model (shown in Supplementary Materials S5) successfully predicted the desired deformations. To supplement the prediction method, we have added the corresponding descriptions to the main text and the changes are highlighted in yellow on Page 11 Column-2 Ln 10-22 and Supplementary Materials S22.

Figure R1-16. Actuation prediction of self-merged ETACs. (A-C) Different self-merged structures with various magnetization profiles. (D-F) Numerical simulation results showing the desired deformations.

[Comment 6]

Minor issues:

Why is it necessary to preheat to 50°C for 30 minutes before cooling to room temperature in Figures 2G-H? What is the temperature and state of the continuum robot during the experiment? These aspects, which may lead to ambiguity, require a more detailed discussion for clarity.

[Response 6]

The softening effect by preheating is an interesting and unique phenomenon observed during ETAC testing and therefore utilized to further enhance the ETACs' functionality.

Given the phase change effect, ETAM respectively experiences reversible softening and hardening effects by heating and cooling within a few seconds to minutes (as shown in the top part of Figure R1-17). We defined the solidified, softened, and fluidified ETAMs as the elastomer state, softened state, and fluid state, respectively. However, we found certain types (flexible:rigid PCL powders $\geq 2:1$) of our customized ETAMs experienced an intermediate state between the elastomer state and softened state. We call these materials 4-states-ETAMs and the other types of ETAMs that only experience elastomer, softened and fluid states are called 3-states-ETAMs. By preheating 4-states-ETAMs to 60°C, the overall thermoplastic base materials can be melted. After cooling the fluidified 4-states-ETAMs to the ambient temperature within a few seconds, the 4-states-ETAMs will stay in the intermediate state for ~10 hours

to cure (Figure R1-17). During this period, the 4-states-ETAMs remain at a stiffness that is higher than the softened stiffness and lower than the elastomer stiffness, showing enhanced flexibility for more delicate deformations.

Back to the Reviewer’s comments, when the terrains are too complex to navigate, it is necessary to conduct preheating to 4-states-ETAMs for enhanced flexibility. During the experiment, ETACs were at ambient temperature and in the intermediate state.

We have added the corresponding description to the main text for clarification and the changes are highlighted in yellow on Page 5 Column-2 Ln 44 to Page 7 Column-1 Ln 14 and Supplementary Materials S23.

Figure R1-17. The phase transition process of 3-states ETAMs and 4-states-ETAMs.

[Comment 7]

During the manufacturing process, permanent magnets are placed beneath the mold to attract the ETAMs in the P and N regions to expel air bubbles. However, what is the approach for the P region, which does not contain magnetic particles?

[Response 7]

We appreciate the Reviewer’s careful comment on the fabrication process. To the best of our knowledge and experience in printing PCL parts, the commercially obtained PCL printing materials show less chance of generating obvious air bubbles during the printing process. We can also prepare our raw PCL into printable materials using industrial processes or outsourcing services. The idea of using permanent magnets placed beneath the mold is based on the slight possibility that tiny bubbles existing around microparticles could expand during the printing process.

[Comment 8]

How is the molten PCL uniformly mixed with the magnetic particles? When the viscosity is too high, it is challenging to achieve an even dispersion of the microparticles through stirring, while if the viscosity is

too low, the microparticles tend to settle rapidly. Is it necessary to adjust the temperature of the PCL to achieve the optimal viscosity for effective mixing with the magnetic particles?

[Response 8]

We understand the Reviewer's concern about the microparticle dispersion issue and would like to provide a detailed explanation as follows:

Different from low-viscosity liquids, such as water/oil, the PCL materials we chose to fabricate ETAMs show moderate viscosity in the fluid state (160-300 Pa·s). To uniformly mix magnetic particles, we first melt the raw PCL at a relatively high temperature (160-200 °C) to achieve a low-viscosity liquid state. At this stage, microparticles can be well dispersed in the melted PCL by stirring at 120 r/min for 5 minutes. By keeping the same stirring speed, the temperature is then adjusted lower to reach the softened state. After an additional 5 minutes of stirring, the mixture is quickly cooled to ambient temperature by placing the culture dish into cold water. It should be noted that the first round of mixing takes advantage of PCL's relatively low viscosity under high temperatures, while the second round of mixing can guarantee the even dispersion of the microparticles due to ETAM's moderate viscosity.

However, no obvious performance errors were observed when comparing ETACs fabricated by the above steps to those fabricated by single-time stirring (only stirring under high temperature). This result indicates that the viscosity range of the melted PCL provides applicable dispersion capability while keeping the dispersed particles unsettled.

We have added the corresponding descriptions to the Method section to help readers better understand. The changes are highlighted in yellow on Page 19 Column-2 Ln 4 to 22.

[References for Reviewer #1]

- [R1] Ojha, A. K., Rajasekaran, R., Hansda, A. K., Singh, A., Dutta, A., Seesala, V. S., ... & Dhara, S. (2023). Biodegradable multi-layered silk fibroin-PCL stent for the management of cervical atresia: in vitro cytocompatibility and extracellular matrix remodeling in vivo. *ACS Applied Materials & Interfaces*, 15(33), 39099-39116.
- [R2] Buchholz, N., Budia, A., Cruz, J. D. L., Kram, W., Humphreys, O., Reches, M., ... & Soria, F. (2022). Urinary Stent Development and Evaluation Models: In Vitro, Ex Vivo and In Vivo—A European Network of Multidisciplinary Research to Improve Urinary Stents (ENIUS) Initiative. *Polymers*, 14(9), 1641.
- [R3] Bajaj, D., Sachdeva, A., & Deepak, D. (2021). Foreign body aspiration. *Journal of Thoracic Disease*, 13(8), 5159.
- [R4] Varshney, R., Zawawi, F., Shapiro, A., & Lacroix, Y. (2014). Use of an endoscopic urology basket to remove bronchial foreign body in the pediatric population. *International Journal of Pediatric Otorhinolaryngology*, 78(4), 687-689.
- [R5] Wankhede, R. G., Maitra, G., Pal, S., Ghoshal, A., & Mitra, S. (2017). Successful removal of foreign body bronchus using C-arm-guided insertion of Fogarty catheter through plastic bead. *Indian Journal of Critical Care Medicine*, 21(2), 96.
- [R6] Avdievich, N. I. . (2011). Transverse Electromagnetic (TEM) Surface Coils for Extremities. *John Wiley & Sons, Ltd*.
- [R7] Vaughan, J. T., Adriany, G., Snyder, C. J., Tian, J., Thiel, T., Bolinger, L., ... & Ugurbil, K. (2004). Efficient high-frequency body coil for high-field MRI. *Magnetic Resonance in Medicine: An Official Journal of the International Society for Magnetic Resonance in Medicine*, 52(4), 851-859.

- [R8] Murbach, M., Neufeld, E., Kainz, W., Pruessmann, K. P., & Kuster, N. (2014). Whole-body and local RF absorption in human models as a function of anatomy and position within 1.5 T MR body coil. *Magnetic resonance in medicine*, 71(2), 839-845.
- [R9] Black, D. R., & Heynick, L. N. (2003). Radiofrequency (RF) effects on blood cells, cardiac, endocrine, and immunological functions. *Bioelectromagnetics*, 24(S6), S187-S195.
- [R10] Wang, S., Zheng, M., Lou, C., Chen, S., Guo, H., Gao, Y., ... & Shang, P. (2022). Evaluating the biological safety on mice at 16 T static magnetic field with 700 MHz radio-frequency electromagnetic field. *Ecotoxicology and Environmental Safety*, 230, 113125.

Reviewer #2:

The core concept introduced here is self-grafting. This is a really interesting idea shared with self-healing polymer materials and devices but shown here in a unique way for continuum magnetic robots. Splitting of segments is done by thermally softening a region of the device and pulling or twisting it apart. Similarly, the device can be rejoined demonstrating high strength once re-joined. Overall I find the work interesting and unique. Although the feasibility for medical use is not clear to me, I think this work can be inspiring to further development of morphology-changing magnetic devices.

[Comment 1]

I don't understand how the twisting is accomplished in the demonstrations. It seems like it is done by hand in Fig 3E rather than magnetically? In particular, the twisting method seems difficult to do magnetically. The videos zoom in on the division site which hides the mechanism of pulling/twisting/bending in these demos. I see in the branch demo the division is done magnetically with much twisting.

[Response 1]

Figure R2-1. Comparisons between the manual and magnetic division approach. (A) i-ii pulling division, iii-iv bending division, and v twisting division. (B) Application scenarios for manual and magnetic divisions.

We agree with the Reviewer that the demonstration of Supplementary Movie S6 is unclear regarding how forces are applied at both ends. Therefore, we have revised the Supplementary Movie S6 and

provided further descriptions.

As described in the main text, self-divisions can be completed manually (Figure 3E) or magnetically (Figure 3G). Figure R2-1 A presents a comparison of manual and magnetic divisions for the Reviewer's information. The manual approach is simple and direct, while the magnetic approach shows apparent advantages in performing remote division in confined lumens (Figure R2-1 B). The applied forces are marked in the figure.

We note that the magnetic twisting is realizable when the holding force is sufficient by a stronger magnetic field strength. The current 10-100 mT magnetic field cannot support enough fixing support for the twisting division of a single ETAC. The realization of twisting division in the branch system demonstration is because there are two branches under a twisting magnetic field, which can perform satisfactory torque transmission.

Regarding the force application distance at the magnetic tip end, the control distance strongly depends on the magnetic field strength generated by the actuation device. According to our optimized test results (we optimized the permanent control distance for practical uses by improving the permanent magnet strength and increasing ETAC's magnet particle content, resulting in an actuation distance of more than 10 cm from the ETAC tip), the control distance for self-division ranges from 5 cm to 12 cm, which is suitable for various BME application scenarios.

We added the corresponding descriptions in the main text to clarify our statement. The changes are highlighted in yellow in Supplementary Materials S20.

[Comment 2]

The device is able to follow complex lumen networks in a desired pattern to varied targets as seen in Figure 2. This is impressive, although the magnetic field generation for these demonstrations is perhaps not realistic for the placement within the body as the field source is held directly under the workspace (5 mm gap) - unreasonably close to the working space. This allows for the generation of very high field gradients in a localized manner to pull the continuum along the desired trajectory. I don't see this as a fair demonstration, and would like to see how the device can target varied lumens with a field source at a distance outside the torso.

[Response 2]

We agree with the Reviewer's suggestion to present a fair demonstration of ETACs' ability to reach various targets in complex lumens. As shown in Figure R2-2, we have supplemented a complex channel navigation demo realized by a Helmholtz coil. We also optimized the permanent control distance for practical uses by improving the permanent magnet strength and increasing ETAC's magnet particle content, resulting in an actuation distance of more than 10 cm from the ETAC tip (Figure R2-2 A).

We installed the phantom in a Helmholtz coil (Figure R2-2 B) and successfully realized the varied target navigation, which was previously realized by permanent magnet control (Figures R2-2 C-D). It proves that ETACs can be accurately controlled to conduct navigations by both an external permanent magnetic field (gradient) and a uniform electromagnetic magnetic field.

We have added a corresponding description in the main text to help the reader understand. The changes are highlighted in yellow on Page 7 Column-2 Ln 14-18.

Figure R2-2. Comparisons between the permanent magnet control and electromagnetic control. (A) Experimental setup of the external permanent magnetic (EPM) actuated navigation. (B) Experimental setup of the electromagnetic actuated navigation. (C) Previous results of EPM actuated navigation. (D) Results of electromagnetic navigation.

[Comment 3]

The localized heating is accomplished with an RF coil held immediately around the continuum with a distance of only a few cm at most. Could this heating be accomplished within the body distance? The manuscript should also comment on the safety of high power RF field generation within the body. Temperature safety is discussed in a reasonable way, but not field safety.

[Response 3]

We understand the heating distance may still not be practically used for some specific scenarios, such as in the digestive and respiratory systems. We regard this issue as an industrial problem that can be addressed by increasing the size and power of the RF machine [R1].

Here, we provide our rationale regarding the concerns on heating efficiency and resulting field safety in detail. Current laboratory RF equipment is limited by its size and power, which are much lower than existing industrial and medical devices [R1-R3]. For example, the RF frequency of existing medical devices ranges from 128 MHz (3T) to 342 MHz (8T), which is more than 400 times higher than that of our laboratory RF machine [R1]. While providing satisfactory RF efficiency, such industrial RF machines can support relatively long-distance heating. The maximum coil diameter we found is 56 cm, which is acceptable for practical application [R2-R3]. Regarding the field safety consideration, there is existing animal safety evaluation research for extreme RF exposure. Researchers found that 10–12 T RF exposure for a continuous 28 days was relatively safe for mice. It was also reported that 3.5–23.0 T RF exposure for 2 h and 7.0–33.0 T for 1 h did not have severe long-term detrimental effects on mice [R4-R5]. Therefore, we regard the heating efficiency and field safety as a well-proven issue that can support our proposed self-grafting technique, which only requires RF exposure for seconds up to several minutes (e.g, 1-2 mins at most for the thermal ablation demonstration, as shown in Figure 8).

We have added the corresponding descriptions to the main text and Supplementary Materials. The changes are highlighted in yellow on Page 10 Column-2 Ln 14-17 and Supplementary Materials S18.

[Comment 4]

Very long thin tendrils of magnetic polymer are seen drawn out in the branching division video. These are not very visible in the figure 3G and not mentioned in the paper. Will these be acceptable clinically, or how could those be dealt with?

[Response 4]

The phenomenon mentioned by the Reviewer can be effectively avoided by optimizing the division process. The shape of division regions differentiates with the division strategy and heating strategy. We have newly conducted a series of division tests to showcase the control strategy, as shown in Figure R2-3.

As an elastomer, an ETAM can be manually stretched into two segments without heating. In this controlled group (Figure R2-3 A), fracture shrinkage was observed similar to any other thermoplastic materials. When we applied an RF field while magnetically or manually stretching an end of the heated specimen, more obvious fracture shrinkage phenomena were observed due to the overall reduced stiffness of the heated region (Figures R2-3 B-C). To avoid such phenomena, magnetically or manually stretching a specimen that went through a heating-cooling loop can generate an evenly divided profile (Figures R2-3 B-C). This is because the heat concentrates at the center of the heated region, whereas the surrounding materials remain in the elastomer state, which will not generate shrinking performances.

The generation of the very long thin tendrils of magnetic polymer is due to the division strategy not being specially selected. Moreover, we would like to note that this phenomenon is also related to the selection of raw PCL. The higher the proportion of rigid PCL powder was involved, the more likely it is to generate tendrils. Adopting the abovementioned division skills will reduce the undesired effect.

Regarding the clinical adaptability of the tendril existence, although ETAMs' biocompatibility was experimentally proven, we suggest users apply the abovementioned division skills to avoid this phenomenon, especially when the ETAC is applied in the respiratory tract, where external objects cannot be easily removed from the body, unlike in the digestive tract.

We have added the corresponding descriptions to the main text and Supplementary Materials. The changes are highlighted in yellow on Page 11 Column-2 Ln 1-9 and Supplementary Materials S21.

Figure R2-3. Self-division skills to avoid long-thin tendrils. (A) Manually divided segments. (B) Magnetically divided segments after RF heating. (C) Manually divided segments after RF heating. (D) Magnetically divided segments after an RF heating-cooling loop.

[Comment 5]

Graduated stiffness profiles in the flexible segments allows for reduced buckling when the continuum is pushed around corners. This seems like a good design choice.

Grasping using multiple divided and joined arms is very interesting. A dedicated grasping basket may

be superior, but this is an interesting demonstration. Similar with the endoscope-mounted grasper.

[Response 5]

We thank the Reviewer's comments on the varied-stiffness design and grasping demonstrations.

[Comment 6]

I struggled to figure out what the P, N and F regions are. The explanation given needed a while for interpretation and I feel that the figures did not help understand this core concept.

P is the with no magnetic particles, and no heat-response.

N contains permanent magnet particles and are highly flexible. They experience a small amount of magnetic heating.

F is stiffness modulating by soft-magnetic (negligible magnetic coercivity), which respond to RF heating.

[Response 6]

We appreciate the Reviewer's suggestion of providing a reader-friendly conclusion. The summarization in the comment is highly precise and easy to follow, which benefits us in providing a clear introduction to P/F/N regions. Considering the arrangement of Figure 1A is already dense, we provide an explanation at the beginning of the "Structure of ETACs" section for readers' better understanding.

The changes are highlighted in yellow on Page 4 Column-2 Ln14-20.

[Comment 7]

The language used is hard to follow at times. Ultimately I was able to find clarity, but it took sustained effort to read.

[Response 7]

We have carefully checked the grammar using and rephrased the language throughout the main text and Supplementary Materials.

[References for Reviewer #2]

- [R1] Avdievich, N. I. . (2011). Transverse Electromagnetic (TEM) Surface Coils for Extremities. *John Wiley & Sons, Ltd*.
- [R2] Vaughan, J. T., Adriany, G., Snyder, C. J., Tian, J., Thiel, T., Bolinger, L., ... & Ugurbil, K. (2004). Efficient high-frequency body coil for high-field MRI. *Magnetic Resonance in Medicine: An Official Journal of the International Society for Magnetic Resonance in Medicine*, 52(4), 851-859.
- [R3] Murbach, M., Neufeld, E., Kainz, W., Pruessmann, K. P., & Kuster, N. (2014). Whole-body and local RF absorption in human models as a function of anatomy and position within 1.5 T MR body coil. *Magnetic resonance in medicine*, 71(2), 839-845.
- [R4] Black, D. R., & Heynick, L. N. (2003). Radiofrequency (RF) effects on blood cells, cardiac, endocrine, and immunological functions. *Bioelectromagnetics*, 24(S6), S187-S195.
- [R5] Wang, S., Zheng, M., Lou, C., Chen, S., Guo, H., Gao, Y., ... & Shang, P. (2022). Evaluating the biological safety on mice at 16 T static magnetic field with 700 MHz radio-frequency electromagnetic field. *Ecotoxicology and Environmental Safety*, 230, 113125.

Dear Editor and Reviewers,

We thank the editor and reviewers again for their valuable, constructive, and critical comments. We have addressed all of the comments in the revised version of the manuscript. A point-by-point response to the concerns and suggestions is provided here.

Reviewer #1

[Comment 1]

The authors have adequately addressed most of the previous comments. However, two major concerns remain unresolved.

First, the term "self-graftable" appears to be misleading. The concept of self-graftability should encompass not only self-division and self-mergence but also inherent mobility characteristics—specifically, the ability to both separate autonomously and actively move to desired locations before merging with itself. This capability is fundamental to creating new structures and functions. Unfortunately, the manuscript fails to demonstrate such comprehensive grafting behavior. While the authors have included additional experiments showing the formation of a simple gripper, the gripper seems unreliable (moreover, the self-division process in this case requires clarification). In addition, Supplementary Material S21 only demonstrates the alignment of the facets to the facets, and not the alignment of the facets to other locations on the body. The authors should address whether it is feasible to achieve the structure and complete the task illustrated in Figure 6 through the claimed self-graftable properties.

[Response 1]

We thank the reviewer for pointing out the mobility concern, which we believe has significantly improved the revised version of the manuscript. In this revision, we first investigated the separated ETACs' inherent mobility, realizing the active motion to desired locations. Next, using the established control strategy, the separated ETACs (untethered) can align accurately with the ETAC main body (tethered) for precise self-mergence. Therefore, the gripper demo has been updated with a deeper understanding of mobility characteristics and alignment strategy.

The inherent mobility of separated ETACs and alignment strategy

Multiple factors determine the control strategy and alignment of separated ETACs, including the magnetization profile, magnetic particle types, alignment locations, etc. Therefore, we have designed the experiment as shown in Table R1-1.

To eliminate the control instability associated with manual operation, we utilized a robot arm (Kuka LBR Med 7 R800) to manipulate an external permanent magnet according to the planned control strategy. Figure R1-1 illustrates the experimental setup. The distance between the magnet and ETACs ranges from 5~8 cm, which can be increased by using a bigger magnet with stronger magnetic field strength. Compared to Helmholtz coil control, we also note that applying an external permanent magnet provides a more convenient and direct method to generate a gradient field for ETACs' movement. An electromagnet field can also realize the same effect by preprogramming field changes.

Table R1-1. Case table of the experimental design

Case No.	Magnetization direction	Particle type	Alignment location	Magnetization profile alignment
1	 Axial	NdFeB	Face-to-face (F2F)	2			Face-to-face (F2F)	3			Edge-to-edge (E2E)	4			Edge-to-edge (E2E)	5			Point-to-point (P2P)	6	 Radial	NdFeB	Face-to-face (F2F)	7			Face-to-face (F2F)	8			Edge-to-edge (E2E)	9			Edge-to-edge (E2E)	10	Point-to-point (P2P)			
11	 Oblique	NdFeB	Face-to-face (F2F)	12			Face-to-face (F2F)	13			Edge-to-edge (E2E)	14			Edge-to-edge (E2E)	15	Point-to-point (P2P)			
16	 No dir.	Fe ₃ O ₄	Face-to-face (F2F)	17			Edge-to-edge (E2E)	18			Point-to-point (P2P)	19	 Axial	NdFeB vs. Fe ₃ O ₄	Face-to-face (F2F)	20			Edge-to-edge (E2E)	21			Point-to-point (P2P)	

Figure R1-1. Experimental setup

Figure R1-2 shows all experimental results and the corresponding control strategies. We demonstrate how a separate (untethered) ETAC is magnetically guided to align with a tethered ETAC main body. The untethered-tethered ETAC alignment is regarded as the most common scenario for self-mergence. Typically, the main body is tethered and inserted manually or via an insertion mechanism, and a single separated ETAC is preferred rather than multiple separated ETACs for better controllability. Regarding the scenario of multiple separated ETACs aligning with a tethered ETAC, we believe it can be further realized for more specific tasks in future works. In this work, we would like to discuss solely the alignment of a single untethered ETAC with the main body. This approach is sufficient to support all the demonstrations presented in the main text.

Figure R1-2. Experimental results of separated ETACs' mobility characteristics and alignment strategy

The results indicate that there are three basic principles for conducting alignment: 1) self-alignment

by utilizing the same magnetization direction (e.g., Case 1, 4, 8, 11, 13, 19, 20, 21), 2) holding for mergence by utilizing external magnetic fields to counteract repulsion (e.g., Case 2, 3, 9, 12, 14, 16), and 3) neglectable magnetization influence (e.g., Case 5, 6, 7, 10, 15, 17, 18). Each case was conducted at least three times to ensure repeatability.

In situ gripper assembly by self-grafting and alignment strategy

Figure R1-3. In situ gripper assembling by self-grafting and alignment strategy.

Self-grafting endows ETACs with the ability to form new structures in a relatively large space after navigating through bottleneck terrains. With the alignment strategy, the graftable end-effector can be assembled in situ through self-grafting. As shown in Figure R1-3, an ETAC was inserted through the glottis and navigated to the trachea under magnetic fields. The ETAC could accurately respond to the magnetic field for bending and conduct self-division by RF heating. By applying a magnetic field in the opposite direction of the separated ETAC's magnetization profile, the separated ETAC could align well with the second arm. Repeating the above steps two more times resulted in four arms aligning with each other, which can merge together to the P-region by activating the RF field. After cooling the heated region, the graftable end-effector (i.e., gripper) was structured and could be utilized to grasp and lock onto the foreign body for removal.

We have added the corresponding descriptions to the main text and the Supplementary Materials. The changes are highlighted in yellow in the main text on Page 10 Column-2 Ln 15-20, Page 14, and Page 15 Column 1 Ln 1-17.

[Comment 2]

Second, I am more concerned about whether the presence of body fluids or lubricants at the contact interface would impede self-mergence rather than self-division. For example, can the network structure shown in Supplementary Materials S24 be successfully achieved in a liquid environment?

[Response 2]

We understand the reviewer's concern regarding the influence of body fluids or lubricants on conducting self-mergence. Here, we would like to provide a more detailed explanation of why it is feasible to achieve self-mergence with the presence of body fluids or lubricants. Moreover, according to the reviewer's question on the network structure in a liquid environment, we set up a more stringent and extreme

condition to support our claim: the in vivo meshing demonstration should be realized underwater, and the mesh should maintain its structure under external magnetic attractions.

There are two fundamental requirements to achieve self-mergence in a fluid environment: 1) keeping the heat accumulation rate higher than the heat dissipation rate and 2) applying an external force (i.e., magnetic force) to expel fluids from the space between contact interfaces. Regarding the first requirement, the feasibility of self-mergence in the presence of body fluids or lubricants strongly depends on the RF power. According to our test, under a 770 kHz RF field, the heat accumulation rate is higher than the heat dissipation rate. The RF frequency of existing commercial heaters can achieve GHz levels, which can provide more efficient heating with a relatively large heating distance. Regarding the second requirement, applying an external magnetic force during or immediately after RF heating leads to reliable mergence between ETACs or between an ETAC and the environment. It should be noted that the necessity of applying external force depends on the heat dissipation rate, which is determined by the amount of body fluids or lubricants present.

As shown in Figure R1-4, the in vivo meshing underwater was successfully realized. The ETAC was inserted through a bottleneck terrain to a large space that was filled with water. The self-division can be conducted both in the air or underwater (as proven in the first version of the response letter). Under magnetic guidance, the separated ETAC can be guided to the target region. Both ends of the separated ETAC can firmly adhere to the phantom underwater by activating the RF heater and applying a downward magnetic force. The mesh can be constructed by repeating the above step three more times. To validate the structural stability of the mesh, we pumped the water out of the phantom and applied a permanent magnet to attract the structured mesh. The mesh successfully maintained its structure under the magnetic influence. Regarding removing the firmly structured mesh from the target area, we can activate the RF heating field while applying a repelling magnetic field to the mesh. The key principle of this process is to create tiny spaces between the EATC-environment interfaces for refilling fluids.

We have added the corresponding descriptions to the Supplementary Materials S24.

Figure R1-4. In vivo meshing underwater was successfully realized, proving that the presence of the body fluid or lubricants also allows the self-mergence.

Reviewer #2

[Comment 1]

My prior concerns were primarily around the practicality of the proposed method. Overall I still have some significant concerns in that regard, but have been partially convinced. Some further specific concerns:

I still find the division method to be vague in the figure. It seems like a significant weakness of the concept that the splitting by magnetic field is quite limited and the demonstration (for example in Fig 3G) is requiring large bending back-and-forth.

[Response 1]

We thank the reviewer again for pointing out the division issue, to demonstrate the proposed three division methods more clearly, we have reconducted the corresponding experiments which are shown in Figure R2-1.

Figure R2-1. Self-division strategies including: (A) pulling division, (B) bending division, and (C) twisting division.

The newly conducted division processes are also shown in Supplementary Movie S20. The actuation distance between the permanent magnet and ETAC is more than 10 cm, which is consistent with our claim. The self-divisions have been successfully realized as planned. The actuation distance can be further increased by increasing the magnet size and the magnetic nanoparticle proportions of the ETAC.

Although all division strategies were successfully realized, we found that self-division is more complicated to conduct than self-emergence. There are two reasons: 1) compared with the self-emergence that commonly requires a single RF heating field, the self-division requires both heating and external magnetic moment for splitting. 2) The magnetic moment generated by the field is required to be relatively large for division. Moreover, a stable rotational magnetic field with an appropriate rotation speed is required in specific cases, such as twisting division. Regarding the reviewer's concern about the large bending back-and-forth in Figure 3G and its corresponding movie, such a phenomenon was also avoided in the abovementioned demonstration. The large bending back-and-forth was led by the insufficient heating temperature and the relatively low magnetic strength of the utilized small permanent magnet. By increasing the heating power and utilizing a bigger permanent magnet, the large bending back-and-force was avoided, as shown in Figure R2-1 and Supplementary Movie S20.

We believe the newly conducted self-division demonstration can validate the feasibility of our proposed self-grafting concept.

We have added the corresponding descriptions to the main text and the Supplementary Materials-S26. The changes in the main text are highlighted in yellow on Page 19 Column 1 Ln 11-29.

[Comment 2]

For the distance from the external permanent magnet to the lumen navigation experiments, you are now more clear that the distance is at least 10 cm for these demos. This is fine, it would be good to explain how much of the human body could be reached under such conditions.

[Response 2]

We agree with the reviewer that it would be useful to explain which part of the human body could be reached under the actuation distance of at least 10 cm. We have provided a detailed list showing the potential biomedical scenarios that could satisfy such actuation distance, as shown in Table R2-1.

We have added the corresponding description in the main text and Supplementary Materials S27. The changes in the main text are highlighted in yellow from Page 16 Column 2 Ln 47 to Page 18 Column 1 Ln 6.

Table R2-1. Potential biomedical scenarios and their corresponding required actuation distances.

Potential biomedical scenarios	Required actuation distance	Supporting literature
Trachea & bronchi navigation and self-grafting)	Skin-trachea mean distance 9.2 ± 1.9 mm Skin-upper trachea distance ~9.2 mm Skin-lower trachea distance ~30.5 mm	Bermede, O., Sarıcaoğlu, M. C., Baytaş, V., Hasde, A. İ., İnan, M. B., & Akar, A. R. (2021). Percutaneous ultrasound-guided versus bronchoscopy-guided dilatational tracheostomy after median sternotomy: A case-control study. Turkish Journal of Thoracic and Cardiovascular Surgery ,

		29(4), 457.
Endoscopic retrograde cholangiopancreatography (Navigation and self-grafting)	Skin-ampulla of Vater mean distance 65 mm	Peng, D., Tao, W., Cheng, Y., Zou, Y. Y., Qian, K., & Zhang, W. (2020). The Shortest Distance from the Skin to Pancreas and the Lower Sternum Angle can Influence Short-Term Surgical Outcomes of Laparoscopy-Assisted Distal Gastrectomy for Gastric Cancer.
Transurethral procedures (Navigation)	Skin-ureter mean distance 83.2 mm	Shan, C. J., Mazzucchi, E., Payão, F., Gomes, A. C., Baroni, R. H., Torricelli, F. C., ... & Srougi, M. (2014). The skin-to-calyx distance measured by renal ct scan and ultrasound. International braz j urol , 40, 212-219.
Endoscopic submucosal dissection (Navigation and self-grafting)	Skin-stomach mean distance 30.9 mm Skin-stomach maximum distance 52 mm	Kiran, G., Yilmaz, I., Aydin, S. E. R. D. A. R., Sanlikan, F., & Ozkaya, E. (2022). The shortest distance between the skin and the peritoneal cavity is obtained with fascial elevation: a preliminary prospective laparoscopic entry study. Facts, Views & Vision in Obgyn , 14(2), 171.
Small-size animal nasal passage intubation, RF thermal ablation in the gastrointestinal tract, respiratory system navigation	Mean diameter of mice nasal passage 1-2 mm Mean diameter of mice esophagus 1-2 mm Mean diameter of mice trachea 1-2 mm Mean diameter of mice bronchi 1-2 mm	Alvites, R. D., Caseiro, A. R., Pedrosa, S. S., Branquinho, M. E., Varejão, A. S., & Maurício, A. C. (2018). The nasal cavity of the rat and mouse—source of mesenchymal stem cells for treatment of peripheral nerve injury. The Anatomical Record , 301(10), 1678-1689. Jelvehgaran, P., de Bruin, D. M., Khmelinskii, A., Borst, G., Steinberg, J. D., Song, J. Y., ... & van Herk, M. (2019). Optical coherence tomography to detect acute esophageal radiation-induced damage in mice: A validation study. Journal of biophotonics , 12(9), e201800440. Kishimoto, K., & Morimoto, M. (2021). Mammalian tracheal development and reconstruction: insights from in vivo and in vitro studies. Development , 148(13), dev198192.